Transitional evolutionary forms in chasmosaurine ceratopsid dinosaurs: evidence from the Campanian of New Mexico

http://orcid.org/0000-0001-9188-4371 Fowler Denver W. 1 2 df9465@yahoo.co.uk
Freedman Fowler Elizabeth A. 1 2 3
1 Badlands Dinosaur Museum, Dickinson Museum Center , Dickinson, ND , USA
2 Museum of the Rockies, Montana State University , Bozeman, MT , USA
3 Department of Natural Sciences, Dickinson State University , Dickinson, ND , USA
Knoll Fabien
Electronic publication date: 2020 Jun 5
Publication date: 2020
Volume: 8
Electronic Location ID: e9251
Received 2019 Nov 20; Accepted 2020 May 7
Copyright: © 2020 Fowler and Freedman Fowler
Copyright year: 2020
Copyright holder: Fowler and Freedman Fowler
License: This is an open access article distributed under the terms of the Creative Commons Attribution License, which permits unrestricted use, distribution, reproduction and adaptation in any medium and for any purpose provided that it is properly attributed. For attribution, the original author(s), title, publication source (PeerJ) and either DOI or URL of the article must be cited.
License URL: https://creativecommons.org/licenses/by/4.0/

Keywords: Dinosaur, Cretaceous, Pentaceratops, Anchiceratops, Ceratopsidae, Chasmosaurinae, Anagenesis, Diversity

Funding: Jurassic Foundation This work was supported by the Jurassic Foundation, and the Horner Fund. The funders had no role in study design, data collection and analysis, decision to publish, or preparation of the manuscript.

==============================
Three new chasmosaurines from the Kirtland Formation (~75.0–73.4 Ma), New Mexico, form morphological and stratigraphic intermediates between Pentaceratops (~74.7–75 Ma, Fruitland Formation, New Mexico) and Anchiceratops (~72–71 Ma, Horseshoe Canyon Formation, Alberta). The new specimens exhibit gradual enclosure of the parietal embayment that characterizes Pentaceratops, providing support for the phylogenetic hypothesis that Pentaceratops and Anchiceratops are closely related. This stepwise change of morphologic characters observed in chasmosaurine taxa that do not overlap stratigraphically is supportive of evolution by anagenesis. Recently published hypotheses that place Pentaceratops and Anchiceratops into separate clades are not supported. This phylogenetic relationship demonstrates unrestricted movement of large-bodied taxa between hitherto purported northern and southern provinces in the late Campanian, weakening support for the hypothesis of extreme faunal provincialism in the Late Cretaceous Western Interior.

Introduction

Intermediate or “transitional” fossils are an expected product of evolution, and are especially celebrated when they occur within major evolutionary transitions (Anderson & Sues, 2007; Wellnhofer, 2010; Daeschler, Shubin & Jenkins, 2006). However, morphological intermediates also occur within the “normal“ evolution that comprises the majority of the fossil record giving us key insight into evolutionary mode, tempo, and trends, but also providing ancient examples of how organisms respond to changes in their environment (Malmgren, Berggren & Lohmann, 1984; Hull & Norris, 2009; Aze et al., 2011; Pearson & Ezard, 2014; Scannella et al., 2014; Tsai & Fordyce, 2015).

In dinosaurs, recognition of morphologic intermediates is confounded by a typically sparse fossil record, characterized by taxa that may be widely separated in space and time, and often known only from single specimens. Despite this, in the Upper Cretaceous rocks of North America a combination of increasingly intensive sampling and newly refined stratigraphy is beginning to fill in gaps in the dinosaur record. This is revealing hitherto unknown morphotaxa that link previously disparate or misunderstood morphologies, and/or define new “end-members” that extend or emphasize stratigraphic morphological trends, challenging previously held assumptions about the mode and tempo of dinosaur evolution (Horner, Varricchio & Goodwin, 1992; Sampson, 1995; Holmes et al., 2001; Ryan & Russell, 2005; Wu et al., 2007; Currie, Langston & Tanke, 2008; Sullivan & Lucas, 2010; Evans, Witmer & Horner, 2011; Scannella & Fowler, 2014; Scannella et al., 2014).

Central to this emergent understanding are the Ceratopsidae: a North American (although see Xu et al., 2010) clade of Late Cretaceous ornithischian dinosaurs that exhibit famously elaborate cranial display structures (Hatcher, Marsh & Lull, 1907). Differences in size or expression of these various horns, bosses, and parietosquamosal frills are used to diagnose different taxa, with ~63 species historically described within two families (the “short-frilled” Centrosaurinae and “long-frilled” Chasmosaurinae; Lambe, 1915), ~26 of which have been erected in the past 10 years. This explosion of new taxa has led some researchers (Sampson & Loewen, 2010; Sampson et al., 2010) to propose that ceratopsids radiated through the Campanian–Maastrichtian into numerous contemporaneous geographically-restricted species. However, it is becoming clear that differences in cranial morphology are not always representative of (contemporaneous) diversity. Cranial morphology has been shown to change significantly through ontogeny (Sampson, Ryan & Tanke, 1997; Horner & Goodwin, 2006; Currie, Langston & Tanke, 2008; Scannella & Horner, 2010; Mallon et al., 2014), such that many historical taxa are now considered growth stages of previously recognized forms. Furthermore, studies conducted within single depositional basins have shown ceratopsid taxa forming stacked chronospecies that do not overlap in time, demonstrating that cranial morphology evolves rapidly (in as little as a few hundred thousand years), and supporting the hypothesis that much of what has been perceived as diversity might instead represent intermediate morphospecies within evolving anagenetic lineages (Horner, Varricchio & Goodwin, 1992; Holmes et al., 2001; Ryan & Russell, 2005; Mallon et al., 2012; Scannella et al., 2014; Fowler, 2017; Holmes et al., 2020).

Intermediate Campanian chasmosaurine ceratopsids were predicted by Lehman (1998; Fig. S1), who showed successive morphospecies of the Canadian genus Chasmosaurus (Dinosaur Park Formation, Alberta; middle to upper Campanian) with a progressively shallowing embayment of the posterior margin of the parietosquamosal frill. This was contrasted with an opposite trend seen in Pentaceratops sternbergii (Fruitland Formation, New Mexico; upper Campanian) to Anchiceratops ornatus (Horseshoe Canyon Formation, Alberta; lower Maastrichtian), whereupon the midline embayment deepens and eventually closes (Lehman, 1993; Lehman, 1998; Fowler, 2010; Fowler, Scannella & Horner, 2011; Wick & Lehman, 2013). This hypothesis matched the stratigraphic occurrence of taxa known at the time, and is supported by new taxa described since 1998 (Vagaceratops (Chasmosaurus) irvinensis; Kosmoceratops richardsoni; Utahceratops gettyi; and Bravoceratops polyphemus; Holmes et al., 2001; Sampson et al., 2010; Fowler, 2010; Fowler, Scannella & Horner, 2011; Wick & Lehman, 2013; although see Supporting Information 1).

However, a recent phylogenetic analysis of chasmosaurines (Sampson et al., 2010) proposed a starkly different relationship (Fig. S2) where a clade (Vagaceratops + Kosmoceratops) instead formed the sister group to a clade composed of Anchiceratops and all other Maastrichtian chasmosaurines. This is significant as it implies that the clade (Vagaceratops + Kosmoceratops) is more closely related to Anchiceratops than is Pentaceratops (i.e., the opposite to the relationship suggested in Lehman (1998)). Indeed, the poorly known chasmosaurine Coahuilaceratops magnacuerna formed a second successive sister taxon to the (Vagaceratops + Kosmoceratops) + (Anchiceratops) clade, suggesting that Pentaceratops is even more distantly related. Also, a Chasmosaurus clade (C. russelli + C. belli) is recovered as separated from (Vagaceratops + Kosmoceratops) (Sampson et al., 2010), despite Vagaceratops (Chasmosaurus) irvinensis being originally recovered as the most derived member of a Chasmosaurus clade by Holmes et al. (2001), and the existence of morphological intermediates between C. belli and V. irvinensis (e.g., cf. C. belli specimen YPM 2016; Lehman 1998; Campbell et al., 2019). Subsequent analyses by Mallon et al. (2011, 2014; using an altered version of the data matrix from Sampson et al. (2010)) recovered cladograms (Fig. S2) that appear “upside down”, with the early Maastrichtian taxa Anchiceratops and Arrhinoceratops occurring in a basal polytomy, and some of the stratigraphically oldest taxa forming the most derived clade (middle to late Campanian (Chasmosaurus belli + Chasmosaurus russelli)); a configuration that would require considerable ghost lineages for many clades. Mallon et al. (2014) acknowledged their unlikely topology, stating that “while the monophyly of the Chasmosaurinae is secure, its basic structure is currently in a state of flux and requires further attention”. This can only be resolved by a combination of character reanalysis and the discovery of new specimens intermediate in morphology between currently recognized taxa.

Here we describe new chasmosaurine material from the Kirtland Formation of New Mexico that forms stratigraphic and morphologic intermediates between Pentaceratops and Anchiceratops. This includes new taxa Navajoceratops sullivani and Terminocavus sealeyi which, although based on fragmentary specimens, both include the diagnostic posterior border of the parietal. Geometric morphometric analysis supports the hypothesis that the posterior embayment of the parietal deepens and closes in on itself over ~2 million years, and that Vagaceratops and Kosmoceratops probably represent the most derived and successively youngest members of a Chasmosaurus lineage. Phylogenetic analysis is less conclusive, but recovers Navajoceratops and Terminocavus as successive stem taxa leading to Anchiceratops and more derived chasmosaurines, and suggests a deep split within Chasmosaurinae that occurs before the middle Campanian. This is supportive of true speciation by vicariance occurring relatively basally within Chasmosaurinae, followed by more prolonged periods of anagenetic (unbranching) evolution. Recent hypotheses of basinal-scale faunal endemism are not supported; however, it appears likely that continental-scale latitudinal faunal variation occurred in the Campanian. The new specimens document incipient paedomorphic trends that come to characterize more derived chasmosaurines in the Maastrichtian, such as Triceratops.

Anatomical abbreviations used in text

Ep, epiparietal numbered from 1 to 3 (e.g., ep1) from medial to lateral; es, episquamosal.

Geological Setting, Materials and Methods

Geological Setting

All newly described material was collected from the upper Campanian Fruitland and Kirtland Formations of the San Juan Basin, New Mexico (Figs. 1 and 2; further information in Supporting Information 1).

Figure 1 Geological map of the southeast San Juan Basin showing localities of radiometric dates and important fossil specimens mentioned in the text.

Collection localities; (A) SMP VP-1500, Navajoceratops sullivani, holotype; (B) NMMNH P-27486, Terminocavus sealeyi, holotype; (C) NMMNH P-33906, Denazin chasmosaurine; (D) NMMNH P-37880, c.f. Pentaceratops sternbergii, parietal fragment; (E) UKVP 16100, c.f. P. sternbergii, complete skull; (F) MNA Pl.1747, c.f. P. sternbergii, complete skull; (G) USNM 8604, Chasmosaurinae sp. anterior end of a parietal median bar; (H) purported collection area of AMNH 6325, P. sternbergii, holotype. (I) NMMNH P-50000, Chasmosaurinae sp., skull missing frill. Radiometric dates recalibrated from Fassett & Steiner (1997) by Fowler (2017). Bedrock geology altered from O’Sullivan & Beikman (1963).

Figure 2 Generalized stratigraphic column of Fruitland and Kirtland Formations with radiometric dates and fossil occurrences.

Specimens mentioned in the main text or supporting information: Pentaceratops sternbergii holotype, AMNH 6325; cf. P. sternbergii, AMNH 1624, 1625; aff. Pentaceratops n. sp., MNA Pl.1747, UKVP 16100, NMMNH P-37880; Navajoceratops sullivani holotype SMP VP-1500; Terminocavus sealeyi holotype, NMMNH P-27468; Chasmosaurinae sp., NMMNH P-50000; “Taxon C”, NMMNH P-33906. Radiometric dates (*) recalibrated from Fassett & Steiner (1997) by Fowler (2017).

Fossil Materials and accepted taxonomy

In order to make proper comparisons, it is necessary to review the taxonomy, stratigraphy, and morphology of historical and type specimens of Pentaceratops and related chasmosaurines. This is discussed in greater detail in Supporting Information 1, and only the following summary is provided here.

One of the problems facing any new analysis of Pentaceratops sternbergii is that although the holotype (AMNH 6325; Osborn, 1923) is a mostly complete skull, it unfortunately lacks the diagnostic posterior end of the parietal, making it difficult to reliably refer other specimens to the taxon. However, the taxonomic importance of the posterior bar was not strongly emphasized until the current work, so many specimens have been historically referred to P. sternbergii by other researchers. Therefore, we have reviewed whether such referrals are appropriate, and consequently revised the referrals of many specimens, while simultaneously attempting to preserve some semblance of taxonomic stability (especially regarding the original material). Some specimens are currently under study by other workers (J. Fry, S.G. Lucas, H.N. Woodward, 2015, personal communication), and so new names are not yet erected. In summary, we follow Lull (1933) and all subsequent workers in considering AMNH 1624 and AMNH 1625 as specimens of cf. P. sternbergii. However, referred specimens MNA Pl.1747 and KUVP 16100 are moved into aff. P. n. sp. along with the new specimen NMMNH P-37880. Partial skull SDMNH 43470 is referred to aff. P. sp., due to uncertainty concerning the relationship of its stratigraphic position and immature ontogenetic condition to morphology. Many other fragmentary specimens previously referred to P. sternbergii (e.g., AMNH 1622) are not considered diagnostic and so are here considered Chasmosaurinae indet. We follow Lehman (1998, the original description) in considering the large skull and skeleton OMNH 10165 as aff. Pentaceratops sp., and not the new taxon Titanoceratops ouranos (Longrich, 2011). Autapomorphies used to diagnose the new taxon Pentaceratops aquilonius (Longrich, 2014) are invalid (Mallon et al., 2016), and it should be considered a nomen dubium.

Concerning other chasmosaurines (for full discussion, see Supporting Information 1), we follow Maidment & Barrett (2011) and Mallon et al. (2012) in considering Mojoceratops perifania (Dinosaur Park Formation, Alberta; Longrich, 2010) as a junior synonym of Chasmosaurus russelli. However, the taxonomy of C. russelli has its own priority problems and as such, specimens will be referred to as "Chasmosaurus russelli" and specimen numbers given. A revision of the epiparietal numbering system (see Supporting Information 1) is used for Chasmosaurus, Vagaceratops (Dinosaur Park Formation, Alberta; Holmes et al., 2001; Sampson et al., 2010) and Kosmoceratops (Kaiparowits Formation, Utah; Sampson et al., 2010), based on comparisons within the Chasmosaurus clade. Bravoceratops polyphemus (Javelina Formation, Texas; Wick & Lehman, 2013) is shown to be a nomen dubium as the element identified as the posterior end of the parietal median bar is reidentified as the anterior end and is shown to be undiagnostic.

The electronic version of this article in Portable Document Format (PDF) will represent a published work according to the International Commission on Zoological Nomenclature (ICZN), and hence the new names contained in the electronic version are effectively published under that Code from the electronic edition alone. This published work and the nomenclatural acts it contains have been registered in ZooBank, the online registration system for the ICZN. The ZooBank LSIDs (Life Science Identifiers) can be resolved and the associated information viewed through any standard web browser by appending the LSID to the prefix http://zoobank.org/. The LSID for this publication is: urn:lsid:zoobank.org:pub:E2ECA33C-63A8-4EFF-9EB4-BCF7ED28C63E. The online version of this work is archived and available from the following digital repositories: PeerJ, PubMed Central and CLOCKSS.

Phylogenetic analysis

Phylogenetic analysis was conducted using an adapted version of the character matrix from Mallon et al. (2014). Edits were made to 22 characters; four new characters were added, making a total of 156 characters (see Supporting Information 2 for further details).

Morphometric analysis

Landmark-based geometric morphometric analysis was used to compare parietal shape among 19 specimens (~9 taxa) of chasmosaurine ceratopsids. The analysis was performed by the software package “Geomorph” (version 2.1.1; Adams & Otárola-Castillo, 2013) within the R language and environment for statistical computing, version 3.1.2 for Mac OSX (http://www.R-project.org/; R Core Team, 2014). 16 landmarks were plotted onto both left and right sides of an image of the parietal in dorsal view. Images used were a combination of photographs and specimen drawings, most of which were taken directly from the literature. Landmarks were specifically selected to represent morphological features that are observed to vary between specimens (Fig. 3). Both left and right sides of the parietal are treated as if they were left sided by mirroring the landmark data for the the right side. This allowed the inclusion of incomplete specimens, or specimens that were not symmetrical.

Figure 3 Morphological landmarks used in morphometric analysis of chasmosaurine parietals.

All landmarks were measured on the parietal only. Points 1 and 2 are the same for both left and right sides, but all other points were mirrored for the right side and analysed along with the non-mirrored left side. Points are defined as follows: (1–4; green): (1) maximum constriction of the median bar, positioned on the midline; (2) posteriormost point of the parietal at the midline ; (3) posteriormost point of the parietal anywhere along the posterior margin; (4) lateralmost point of the parietal ; (5, yellow): (5) point at which the lateral ramus of the posterior bar meets the median bar as expressed on the posteriomedial border of the parietal fenestra, may be marked by a change in angle of the fenestra border; (6, 7; magenta): (6) posteriormost point of parietal fenestra ; (7) lateralmost point of parietal fenestra; (8–13; blue): (8) contact point of the medial margin of epiparietal 1 with the parietal itself; (9) contact point of the lateral margin of epiparietal 1 with the parietal itself; (10) contact point of the medial margin of epiparietal 2 with the parietal itself; (11) contact point of the lateral margin of epiparietal 2 with the parietal itself; (12) contact point of the medial margin of epiparietal 3 with the parietal itself; (13) contact point of the lateral margin of epiparietal 3 with the parietal itself; (14–16; red): (14) the contact point of the midpoint of epiparietal 1 with the parietal itself; (15) the contact point of the midpoint of epiparietal 2 with the parietal itself; (16) the contact point of the midpoint of epiparietal 3 with the parietal itself. Colors are intended to aid in visual distinction only. Points illustrated on “Chasmosaurus russelli” referred specimen CMN 2280, adapted from Godfrey & Holmes (1995).

Although the parietal of Agujaceratops mariscalensis (UTEP P.37.7.065, 070, 071) is fragmentary, the reconstruction of Lehman (1989) is included for comparison, although only the left side was analysed since it is only this side that is based on fossil material. Only the right sides of Kosmoceratops richardsoni holotype UMNH VP 17000 and “Chasmosaurus russelli” referred specimen TMP 1983.25.1 were analysed as the left sides were damaged and missing critical areas. Only the left side of Chasmosaurus belli specimen AMNH 5402 was used as the right side is unusually distorted.

Landmarks were digitized within the R program using “digitize2d” (version 2.1.1; Adams & Otárola-Castillo, 2013). Parietals were rotated and scaled using Generalized Procrustes Analysis (using the function “gpagen”) so that shape was the only difference among specimens. Consequent Procrustes coordinates were analyzed in a Principal Components Analysis (function “plotTangentSpace”).

Results

Systematic Paleontology

DINOSAURIA Owen, 1842, sensu Padian & May, 1993

ORNITHISCHIA Seeley, 1887, sensu Sereno, 1998

CERATOPSIA Marsh, 1890, sensu Dodson, 1997

CERATOPSIDAE Marsh, 1888, sensu Sereno, 1998

CHASMOSAURINAE Lambe, 1915, sensu Dodson, Forster & Sampson, 2004

Pentaceratops sternbergii (Osborn, 1923)

Type specimen - AMNH 6325 (Osborn, 1923), nearly complete skull, missing the mandible and the posterior half of the parietal and squamosals.

Referred specimens - AMNH 1624, nearly complete skull, missing mandible and the medial part of the parietal; AMNH 1625, nearly complete frill, missing anterior end of the parietal and right squamosal, and most of the left squamosal. Referred to as cf. Pentaceratops sternbergii.

Locality and Stratigraphy - AMNH 6325, 1624, and 1625 were all collected by C.H. Sternberg in 1922 and 1923 from the Fruitland Formation, San Juan Basin, New Mexico (Figs. 1 and 2; see Supporting Information 1 for discussion).

Diagnosis - Chasmosaurine ceratopsid characterized by the following combination of characters (modified from Lehman, 1998; and Longrich, 2014): Posterior bar of the parietal M-shaped, with well-developed median embayment. Arches of the M-shape angular, with apex of arch occurring at locus ep2. Anteroposterior thickness of the parietal posterior bar uniform (or nearly so) from medial to lateral. Three large subtriangular epiparietals. Ep1 curved dorsally or anterodorsally and sometimes twisted such that the base of the epiparietal contacts the posterior margin of the frill laterally, and lies on the dorsal surface of the frill medially. Parietal median bar with slender ovoid cross section. Frill long and narrow, broader anteriorly than posteriorly. Posteriormost episquamosal enlarged relative to penultimate episquamosal. Parietal fenestrae subangular in shape. Postorbital horns present and relatively slender, curving anteriorly (at least in adults). Epijugal spikelike, more elongate than in other chasmosaurines, curving ventrally. Nasal horn positioned over the naris.

Can be distinguished from Chasmosaurus by the following characters: Lateral rami of the parietal posterior bar meet medially at <90°, rather than >90° (although one specimen of "C. russelli", CMN 8803 bears an angle of 87°). Ep1 occurs within the embayment of the parietal posterior bar, rather than at the lateral edges of the embayment ("C. russelli") or as an elongate ridge occupying most of the lateral ramus (C. belli / sp.). Ep1 typically curved anteriorly and oriented anterolaterally, rather than pointing posteriorly. Ep2 oriented to point posteriorly rather than posterolaterally. Ep2 triangular and symmetrical (or nearly so) rather than asymmetrical. Posteriormost point of the parietal posterior bar (apex of the curved lateral ramus) occurs at locus ep2 rather than ep1. Maximum point of constriction for the parietal median bar occurs approximately halfway along its length, rather than within the posterior third. Frill broader anteriorly than posteriorly. Nasal horn positioned over the naris rather than 50% or more positioned posterior to the naris. Premaxillary flange restricted to dorsal margin of premaxilla, rather than along entire anterior margin of external naris. Postorbital horns elongate and anteriorly curved (in large individuals assumed to represent adults), rather than abbreviated, resorbed, and/or curved posteriorly (adapted from Forster et al., 1993; Maidment & Barrett, 2011; Longrich, 2014).

Can be distinguished from Utahceratops gettyi by the following characters: nasal horn more anterior than U. gettyi, being positioned over the naris rather than posterior to the naris. Postorbital horns elongate and anteriorly oriented (in large individuals assumed to represent adults), rather than abbreviated or resorbed and oriented anterolaterally.

Comment - The virtually complete parietosquamosal frill, AMNH 1625 is the most diagnostic of the original referred materials. As AMNH 1624 is missing the central part of the parietal it can only be tentatively referred to the same taxon as AMNH 1625 based on the following shared diagnostic characters (which are not seen in aff. Pentaceratops n. sp. specimens; MNA Pl. 1747, KUVP 16100, and NMMNH P-37880): the posteriormost point of the parietal posterior bar is positioned at locus ep2. Ep2 is not positioned within the parietal median embayment. Ep2 is oriented posteriorly. The lateralmost edge of the lateral rami of the parietal posterior bar is slightly expanded in AMNH 1624, more so than in AMNH 1625, but less so than seen in MNA Pl.1747 and KUVP 16100. The M-shape of the posterior bar is slightly angular in AMNH 1624, more similar to AMNH 1625 than the rounded M-shape in MNA Pl.1747 and KUVP 16100.

Both AMNH 1624 and 1625 were referred to Pentaceratops sternbergii without comment by Lull (1933; see Supporting Information 1). From 1933 to 1981, the defined morphology of P. sternbergii was based on the combination of these specimens along with the holotype AMNH 6325, thus forming a hypodigm (Simpson, 1940). Rowe, Colbert & Nations (1981) referred the then newly discovered MNA Pl.1747 and KUVP 16100 to P. sternbergii, but implicitly recognized that these new specimens were distinct from the P. sternbergii hypodigm. They state (p. 40) that the reconstructed frills of AMNH 6325 and 1624 were “on the basis of (MNA Pl.1747), seen to be incorrect”. The frills of AMNH 6325 and 1624 were presumably reconstructed based on the complete frill, AMNH 1625 (which Rowe, Colbert & Nations (1981) acknowledge the extistence of, but had not been able to locate, nor observe a photograph). Following this, based on the morphology of the posterior end of the parietal, here we show that MNA Pl.1747 and KUVP 16100 should be referred to a different taxon from AMNH 1624 and 1625.

The P. sternbergii holotype specimen AMNH 6325 lacks the diagnostic posterior bar of the parietal, so we cannot currently know whether the holotype would have been more similar to AMNH 1624 and 1625, MNA Pl.1747 and KUVP 16100, or a different morphology entirely. A possible exception is that the preserved portion of the parietal median bar of AMNH 6325 is narrow and particularly elongate, more so than the median bars of chasmosaurines recovered from the Kirtland Formation (Navajoceratops, Terminocavus, new taxon C, and “Pentaceratops fenestratus”). AMNH 6325, 1625, and 1624, MNA Pl.1747, and KUVP 16100 are all recorded as having been collected in the Fruitland Formation (with no better stratigraphic resolution available for the AMNH specimens; see Supporting Information 1), so that stratigraphy is mostly uninformative regarding their potential separation.

Despite the inadequacy of the holotype AMNH 6325, it is desirable to conserve the name Pentaceratops, and P. sternbergii. In order to do so the original hypodigm of Lull (1933) is maintained here, and we thus refer specimens AMNH 1624 and 1625 to cf. P. sternbergii. For this to be formalized, it would be best to petition the ICZN to transfer the holotype to another specimen, preferably AMNH 1625. Without transfer of the holotype, Pentaceratops and P. sternbergii should be considered nomen dubia, and a new taxon erected for diagnostic specimen AMNH 1625 and (possibly) 1624.

aff. Pentaceratops n. sp.

Referred specimens - MNA Pl.1747, complete skull and partial postcranium; KUVP 16100, complete skull; NMMNH P-37880, partial right lateral ramus of parietal posterior bar.

Locality and Stratigraphy - All specimens were collected from the upper part of the Fruitland Formation, San Juan Basin, New Mexico (Figs. 1 and 2; see Supporting Information 1).

Diagnosis - Differs from cf. Pentaceratops sternbergii (principally, AMNH 1625) by possession of the following characters. Arches of the M-shaped parietal posterior bar rounded rather than angular. Apices of M-shaped arch more laterally positioned, occurring either between loci ep2 and ep3, or at locus ep3, rather than at locus ep2. Lateral rami of the parietal posterior bar become more anteroposteriorly broad from medial to lateral, rather than being “strap-like” with near-uniform thickness. Locus ep2 positioned on the lateralmost edge within the embayment, oriented medioposteriorly. Lateral parietal bars more strongly developed.

Comment - KUVP 16100 and MNA Pl.1747 have historically been referred to Pentaceratops sternbergii (Rowe, Colbert & Nations, 1981; Lehman, 1993, 1998; Longrich, 2011, 2014), but are here shown to differ from the historical hypodigm (Lull, 1933; see above). NMMNH P-37880 is described for the first time in Supporting Information 1.

Morphological features known to indicate relative maturity in chasmosaurines (Horner & Goodwin, 2006, 2008) suggest that referred specimens of aff. Pentaceratops n. sp. are not fully mature (MNA Pl.1747, subadult or adult; KUVP 16100, subadult; and NMMNH P-37880, subadult; see Supporting Information 1). Since AMNH 1625 exhibits features supportive of full adult status (see Supporting Information 1), then this raises the possibility that any morphological differences between cf. P. sternbergii and aff. Pentaceratops n. sp. are ontogenetic rather than taxonomic. This is possibly supported by stratigraphic data as AMNH 1625 is thought to have been collected from below the Bisti Bed sandstone, as were MNA Pl. 1747, KUVP 16100, and NMMNH P-37880. However, given the close similarity in size and ontogenetic status of AMNH 1625 and MNA Pl.1747, we prefer to consider their morphological differences as taxonomic, although remain open to the ontogenetic hypothesis. Further discovery of mature material with stratigraphic data would help resolve this question.

Navajoceratops sullivani gen. et sp. nov.

urn:lsid:zoobank.org:act:038D3DF1-DB41-48AF-9791-14C846971133

Etymology - Navajoceratops, “Navajo horned face”, after the Navajo people indigenous to the San Juan Basin; sullivani, after Dr. Robert M. Sullivan, leader of the SMP expeditions to the San Juan Basin that recovered the holotype.

Holotype - SMP VP-1500; parietal, squamosal fragments, fused jugal-epijugal, other unidentified cranial fragments. Collected in 2002 by Robert M. Sullivan, Denver W. Fowler, Justin A. Spielmann, and Arjan Boere.

Locality and Stratigraphy - SMP VP-1500 was collected from a medium brown-grey mudstone at SMP locality 281 (“Denver’s Blowout”), Ahshislepah Wash, San Juan Basin, New Mexico (Sullivan, 2006; detailed locality data available on request from NMMNH). The locality occurs in the lower part of the Hunter Wash Member of the Kirtland Formation (Fig. 2), ~43 m stratigraphically above the uppermost local coal, and ~6 m stratigraphically above the top of a prominent sandstone once thought to represent the Bisti Bed (SMP locality 396; “Bob’s Bloody Bluff”; Sullivan, 2006), but now thought to be ~4 m above the Bisti Bed (R.M. Sullivan, 2020, personal communication; see Supporting Information 1). Hence SMP VP-1500 occurs stratigraphically higher than specimens referred to cf. Pentaceratops sternbergii and aff. Pentaceratops n. sp. which all occur below the Bisti Bed sandstone.

Most elements of SMP VP-1500 were collected as weathered surface material, with the exception of the parietal, which was only partly exposed and required excavation. The parietal was preserved dorsal-side up with the median bar broken and displaced ~10 cm anteriorly (see Fig. S4), and the distal part of the right ramus of the posterior bar broken and displaced ~20 cm posterolaterally.

Diagnosis - Can be distinguished from aff. Pentaceratops n. sp. by the following characters: Lateral rami of the parietal posterior bar meet medially at a more acute angle (~60°, rather than 87 or 88°; KUVP 16100, MNA Pl.1747, respectively). Median embayment of the parietal posterior bar especially deep, extending anterior to the posteriormost extent of the parietal fenestrae (which consequently overlap anteroposteriorly slightly with ep2).

Description

Parietal - The parietal (Fig. 4) is missing the lateral bars and most of the anterior end, but is otherwise relatively complete. Deep vascular canals are visible across the dorsal and ventral surfaces, and are especially well developed on the ventral surface. The posterior and medial borders of both parietal fenestrae are well preserved; enclosing the parietal fenestrae that are large and subangular. Six epiparietal loci are interpreted to occur on the posterior bar, numbered ep1–3 on each side.

Figure 4 Navajoceratops sullivani holotype SMP VP-1500 parietal.

Dorsal (A) and ventral (B) views. Cross section of median bar (mb) illustrated on dorsal view. Ep1 mostly removed during extraction or preparation (see Fig. S4 for original extent). em, median embayment of the posterior bar; ep, epiparietal loci numbered by hypothesized position (no epiossifications are fused to this specimen); f, parietal fenestra; L-lr/R-lr, Left/Right lateral rami of the posterior bar; te, tapering lateral edges of the median bar. Scalebar = 10 cm. Reconstruction adapted from Lehman (1998).

The preserved portion (~60%) of the median bar measures 37.4 cm in length, and tapers anteriorly, measuring 4.1 cm wide at the anteriormost end. The dorsal and ventral surfaces of the median bar are convex, with lateral margins of the median bar tapering to give a lenticular cross section. These tapering lateral edges broaden posteriorly. The dorsal surface bears no prominent medial crest, ridge, or bumps (such features are restricted to the anteriormost third of the median bar in other chasmosaurines; e.g., Anchiceratops, Brown, 1914, Mallon et al., 2011; “Torosaurus” utahensis, Gilmore, 1946; “Torosaurus” sp., Lawson, 1976; “Titanoceratops”, Longrich, 2011; Triceratops, Hatcher, Marsh & Lull, 1907; see discussion in Supporting Information 1 on “Bravoceratops”, Wick & Lehman, 2013). Two fragments found during excavation may represent parts of the anterior end of the median bar. The largest fragment bears parallel vascular traces along its length, suggesting it is indeed part of the midline of the anterior end of the parietal.

The median bar and lateral rami of the posterior bar form a Y-shape, with the rami of the posterior bar meeting at an angle of 60°, forming a deep U-shaped median embayment that incises 13.2 cm anterior to the posteriormost extent of the parietal fenestrae. The lateral rami are slightly wavy rather than straight, and form an M-shape with the curved apices of the M occurring between epiparietal loci ep2 and ep3. The lateral rami of the posterior bar vary in anteroposterior thickness, being relatively thick at the contact with the median bar (R: 11.5 cm; L: 12.8 cm), reaching their narrowest point slightly medial of the apex (R: 9.37 cm; L: 9.17 cm), broadening at the apex (R: 20.2 cm; L: 20.0 cm), then narrowing again laterally towards the contact with the squamosal.

There are two raised areas on either side of the anterodorsal margin of the posteromedial embayment. During excavation, the lateral rami bore an especially thick concretion in this area, suggesting bone underneath the surface (see Fig. S4); however, if present, all of this bone was lost during preparation. A very similar raised area is considered as representing ep1 in Utahceratops referred specimen UMNH VP 16671 (Sampson et al., 2010). This raised area is also considered as an attachment locus of ep1 in aff. Pentaceratops n. sp. specimen KUVP 16100 and aff. P. sternbergii specimen SDNHM 43470, and is occupied by a fused dorsolaterally oriented ep1 in specimens MNA Pl.1747, and the left side of AMNH 1625. Therefore it is tentatively suggested that these raised areas are the attachment loci where a once well-fused ep1 would have resided. Both the left and right ep2 are preserved imperceptibly fused to the posterior bar and project posteromedially into the embayment, almost touching medially. Ep2 on both sides is a rounded D-shape, rather than triangular. There is no evidence of ep3, which might be expected to occur at the lateralmost edges of the lateral rami. However, although ep3 is typically reconstructed as occurring in this position in Pentaceratops sternbergii (e.g., Lehman, 1998), only AMNH 1624 and 1625 actually preserve an ep3, and in these specimens it abuts or straddles the squamosal-parietal margin (although see notes on MNA Pl.1747 in Supporting Information 1). An isolated D-shaped frill epiossification (Fig. S5) was recovered adjacent to the parietal during excavation of SMP VP-1500. It is unlike the spindle-shaped or triangular episquamosals, and so may be an unfused ep1 or ep3.

Squamosal - SMP VP-1500 includes pieces of at least one squamosal (probably a left), but most of these are too small and fragmentary to impart much morphological knowledge. The two largest fragments are shown in Fig. S6. The first fragment (Figs. S6A and S6B) is roughly triangular in shape and preserves part of the lateral margin, which is thicker than the more medial area. Two episquamosals are preserved fused to the lateral margin. Both episquamosals are trapezoidal or D-shaped. The second large fragment (Figs. S6C and S6D) is also triangular, but is narrower than the first fragment and as such might be part of the distal blade of the squamosal. Few features are diagnostic on the second fragment, although a relatively complete straight edge may represent the medial margin where the squamosal articulates with the parietal. Both of the large fragments exhibit the woven, vascularized surface texture typical of ceratopsid skull ornamentation.

Jugal/Epijugal or Episquamosal - A ~10 cm fragment (SMP VP-1813) bearing a pointed epiossification possibly represents the ventral margin of a fully fused right jugal, quadratojugal, and epijugal (Fig. S7). It was collected as float from the same locality as SMP VP-1500 and possibly pertains to the same individual. The epijugal is relatively stout, but not unusually so, nor is it especially long or pointed. An alternative identification of this element is a large episquamosal. Regardless, the specimen is not especially diagnostic.

Terminocavus sealeyi gen. et sp. nov.

urn:lsid:zoobank.org:act:6E5A8D79-1F2C-484F-BED7-7C556C5C062A

Etymology - Terminocavus, “coming to the end of (or “last stop for”, as in a train terminus) the cavity” after the nearly-closed parietal embayment; sealeyi after Paul Sealey who discovered the holotype specimen.

Holotype - NMMNH P-27468; parietal, partial squamosal, jugal, epijugal, partial quadratojugal, partial sacrum, vertebral fragments. Collected in 1997 by Paul Sealey.

Locality and Stratigraphy - NMMNH P-27468 was collected from a grey siltstone beneath a white channel sandstone (locality NMMNH L-3503; precise locality data available from NMMNH upon request) in the middle of the Hunter Wash Member, stratigraphically intermediate between ash 2 (75.02 ± 0.13 Ma) and ash 4 (74.57 ± 0.62) (Fowler, 2017). Although in Fig. 1 NMMNH L-3503 appears to be approximately halfway between these radiometrically dated horizons, it occurs in a topographic high between Hunter Wash and Alamo Wash, placing it stratigraphically closer to ash 4. Trigonometric calculations place the locality at ~83 m stratigraphically above ash 2, and ~48 m stratigraphically below ash 4 (based on a northeast dip of 1°). This agrees quite well with Bauer (1916) who published a thickness of 1,031 feet (314 m) for the Hunter Wash Member (then called the Lower Shale Member) at Hunter Wash itself. However, in their description of the ashes, Fassett & Steiner (1997) suggest that the ashes are separated stratigraphically by only ~45 m. This would appear to be an underestimate, based on both Bauer (1916) and on the fact that ash 4 is ~130 ft (40 m) topographically higher than ash 2, and ~5 km NE (basinwards, parallel to 1–3° dip).

It is worth mentioning that the locality is only ~0.6 km SE of another ash (JKR-54) that was dated by Brookins & Rigby (1987). The large margin of error for their K/Ar date of 74.4 ± 2.6 Ma (sanidine) places it within the expected range based on the more precise Ar/Ar recalibrated dates of Fassett & Steiner (1997; recalibrations by Fowler (2017)). Although the K/Ar date of Brookins & Rigby (1987) is imprecise and not really usable, the JKR-54 horizon would be useful to resample in future San Juan Basin research.

Comment - NMMNH P-27468 has only previously been mentioned in an abstract by Sealey, Smith & Williamson (2005) where it was identified as an aberrant specimen of Pentaceratops sternbergii. NMMNH P-27468 is the only diagnostic chasmosaurine specimen from the middle or upper part of the Hunter Wash Member of the Kirtland Formation; other Kirtland Formation chasmosaurine specimens collected by C.H. Sternberg in the 1920s (described by Wiman (1930); including the holotype of “Pentaceratops fenestratus”; see Supporting Information 1) are mostly undiagnostic or fragmentary, and lack detailed locality and stratigraphic data.

Diagnosis - Differs from Navajoceratops holotype SMP VP-1500 by the following characters: Posterior bar flattened and plate-like (i.e., not bar-like). Lateral rami of the parietal posterior bar strongly expanded anteroposteriorly both medially and laterally. Maximum anteroposterior thickness of the posterior bar ~35% of the parietal maximum width (compared with <30% in Navajoceratops and ~19–30% in aff. Pentaceratops n. sp.). Median embayment of the posterior bar narrower and more notch-like. Parietal fenestrae subrounded rather than subangular.

Description

Parietal - The parietal of NMMNH P-27468 (Fig. 5) is missing ~50% of the anterior end, but is otherwise relatively complete forming a rounded-M or heart-shape reminiscent of later-occurring chasmosaurines such as the holotype of “Torosaurus gladius” YPM 1831. The parietal is not formed of obvious narrow bars as seen in stratigraphically older chasmosaurines, rather, it is expansive, flat, and more plate-like. The parietal is comparatively thin (typically ~1–2 cm in thickness), although this may reflect postburial compression. Bone surfaces have a thin concretion of sediment that obscures most fine surface detail, although shallow vascular canals are visible on some areas of the dorsal surface. The ventral surface is mostly either obscured by concreted sediment or damaged, but in some places longitudinal vascular canals can be observed, similar to those in Navajoceratops and other chasmosaurines. The posterior and medial borders of both parietal fenestrae are well preserved. However, the posterior, median, and lateral bars are expanded at the expense of the parietal fenestrae, which are thus slightly reduced in size relative to stratigraphically preceding chasmosaurines. The fenestrae are subrounded in shape, comparable to derived chasmosaurines such as Anchiceratops and triceratopsins, but unlike the subangular- or angular-shaped fenestrae of stratigraphically older chasmosaurines.

Figure 5 Terminocavus sealeyi holotype NMMNH P-27468 parietal.

Dorsal (A) and ventral (B) views. Paired ep1 are deflected dorsally. em, median embayment of the posterior bar; ep, epiparietal loci numbered by hypothesized position (no epiossifications are fused to this specimen); f, parietal fenestra; lb, lateral bar; L-lr/R-lr, Left/Right lateral rami of the posterior bar; mb, median bar; te, tapering lateral edges of the median bar. Scalebar = 10 cm. Reconstruction adapted from Lehman (1998).

The preserved portion of the median bar measures 31.1 cm in length and tapers anteriorly. The dorsal surface of the midline bar is convex, lacking a medial crest, ridge, or bump. The ventral surface of the median bar is flat to weakly convex. The lateral margins of the median bar taper to give a lenticular cross section. The median bar bears small flanges that run along the lateral edges, and are directed laterally into the fenestrae. Although broken anteriorly, the flanges are more laterally extensive than in Navajoceratops and other stratigraphically preceding chasmosaurines.

The left and right lateral bars are incomplete and probably represent only ~50% of their original length. The preserved portions are of nearly equal antero-posterior length, and are almost parallel, suggesting the anterior end of the parietal was slightly narrower than the posterior, or at least narrowed in its midline (as in cf. Pentaceratops sternbergii MNA Pl. 1747; Rowe, Colbert & Nations, 1981). Both lateral bars are convex dorsally, and flat to weakly convex ventrally. Dorsoventral thickness decreases laterally such that they are moderately lenticular in cross section. The lateral edges which articulate with the squamosal are thin and plate-like. Each lateral bar bears a relatively large (diameter ~5 mm) anteroposteriorly-oriented blood vessel groove that runs toward to the lateral rami of the posterior bar. However, like other blood vessel traces on this specimen, the grooves are shallow and difficult to trace onto the lateral rami.

The lateral rami of the posterior bar meet medially at an angle of 73°, which is steeper than in stratigraphically preceding chasmosaurines, however, it is awkward to measure as the lateral rami are curved rather than being straight lines (see Supporting Information Figs. S10 and S11). The lateral rami are anteroposteriorly thicker than those of Utahceratops, Pentaceratops, and Navajoceratops, but less so than in Anchiceratops. They vary in anteroposterior thickness from medial to lateral, being at their narrowest medially, at the contact with the median bar (Right: 13.2 cm; Left: 12.2 cm), reaching their broadest point at the apex (Right: 23.4 cm; Left: 23.6 cm), then narrowing again laterally towards the contact with the squamosal.

The median embayment is narrower than in preceding chasmosaurines, forming a notch that is almost enclosed by the first pair of epiparietals. The embayment does not extend anterior to the posteriormost border of the parietal fenestrae. The anterior edge of the embayment is notably thickened, similar to that seen in cf. Utahceratops gettyi specimen UMNH VP-16671 (Sampson et al., 2010). On the left lateral ramus, the thickened border of the embayment is extended continuously in a posterior direction helping form the anteromedial edge of the left ep1 (see below). However, on the right side, the thickened border is discontinuous, forming a small prominent bump below the main part of the ep1. A similar double bump at the ep1 locus is seen on the left side of cf. U. gettyi specimen UMNH VP-16671 where it is labeled as a “dorsal parietal process”, with the right side continuous (Sampson et al., 2010).

Five epiparietals are preserved fused to the parietal (i.e., at least one missing), which is probably representative of three pairs of epiparietals (ep1–3) as is typical for chasmosaurines. The medialmost pair of epiparietals are considered to represent ep1, and are positioned on the medial margin of the median embayment, as it is in specimens referred to cf. Pentaceratops sternbergii, aff. Pentaceratops n. sp., and cf. Utahceratops gettyi. The left ep1 is triangular, whereas the right ep1 was probably also triangular but is missing the distal tip, instead exhibiting a shallow, possibly pathological trough. This is of interest because if the right ep1 tip was present then the epiparietals are close enough (separated by only ~5 mm) that they would probably have touched (especially if they bore keratinous sheaths). Ep1 is the only epiparietal that does not lie flat within the plane of the parietal. Both left and right ep1 are deflected slightly dorsally, similar to the ep1 on the right side of cf. Pentaceratops sternbergii specimen AMNH 1625 and parietal fragments referred to “Pentaceratops aquilonius” (CMN 9814; Longrich, 2014; or Spiclypeus, Mallon et al., 2016; see Supporting Information 1). Ep2 is preserved on both sides, although it is broken slightly on the right side. Ep2 is triangular and projects posteromedially from the posterior bar, laying flat within the plane of the rest of the parietal. Ep3 is only preserved on the left side where it is fused to the posterior bar. There is an empty space at locus ep3 on the right side. Ep3 is more D-shaped than triangular and projects posteriorly laying flat within the plane of the rest of the parietal. There is no indication of an epiparietal more lateral than the ep3 locus, despite there probably being enough space for an additional epiossification (as seen in some specimens of Anchiceratops; Mallon et al., 2011).

Right Squamosal - The preserved right squamosal (Fig. S8) comprises a nearly complete anterior end (including the narrow processes that articulate with the quadrate and exoccipital), the anteriormost episquamosal, and most of the medial margin of the squamosal blade. Almost the entire lateral margin and the posterior end are not preserved. The medial margin is robust and forms the squamosal bar. Although incomplete, the squamosal bar is long enough to suggest that the squamosal itself was elongate, as seen in most adult chasmosaurines, rather than short and broad, as seen in young chasmosaurines (Lehman, 1990; Scannella & Horner, 2010); the preserved portion measures 83 cm in length, and the conservative reconstruction (Fig. S8) is 94 cm. Lateral to the squamosal bar, the squamosal dorsoventrally thins and is broken. The single preserved episquamosal is fused to the anterolateral border and represents the anteriormost episquamosal. It is common in chasmosaurine specimens for the anteriormost episquamosal to be fused to the anterolateral border of the squamosal, suggesting that it is one of the first episquamosals to fuse through ontogeny (Godfrey & Holmes, 1995; Campbell et al., 2016). The episquamosal is very rugose and not obviously triangular in shape.

Jugal/Epijugal - NMMNH P-27468 also has a fused left jugal, epijugal, and quadratojugal (Fig. S9). The orbital margin of the jugal is not preserved, and only a little remains of the anterior process. The ventral part of the jugal is tongue shaped, terminating in the indistinguishably fused epijugal. The epijugal is large and robust, but not notably long. Only the ventralmost part of the quadratojugal is preserved, fused to the epijugal. Similar to the parietal, surface texture is partly obscured by sediment, but some shallow vascular grooves are visible.

Chasmosaurinae sp. “taxon C”

Material - NMMNH P-33906; parietal median bar, epijugal, indeterminate skull fragments, vertebral fragments.

Locality and Stratigraphy - NMMNH P-33906 was collected in 2001 by Thomas E. Williamson at NMMNH locality L-4715, from the De-na-zin Member of the Kirtland Formation at South Mesa, San Juan Basin, New Mexico (Figs. 1 and 2; precise locality coordinates are available from NMMNH). Two radiometrically dated ashes (at Hunter Wash, ~10 km to the northwest) bracket the age of the De-na-zin Member of the Kirtland Formation. Ash H (73.83 ± 0.18 Ma) occurs less than 5 m above the basal contact of the De-na-zin Member with the underlying Farmington Member (Fassett & Steiner, 1997; Sullivan, Lucas & Braman, 2005). Ash J (73.49 ± 0.25 Ma) occurs 4.9 m below the upper contact of the De-na-zin Member with the overlying Ojo Alamo Sandstone (Fassett & Steiner, 1997; both radiometric dates recalibrated by Fowler (2017), from Fassett & Steiner, 1997). NMMNH P-33906 therefore occurs between 73.83 Ma and 73.49 Ma.

Comment - Although fragmentary, the previously undescribed specimen NMMNH P-33906 represents one of the few records of chasmosaurines from the De-na-zin Member of the Kirtland Formation, and preserves the median bar of the parietal, which is diagnostic enough to permit comparison to other chasmosaurines.

Diagnosis - Differs from Utahceratops, cf. Pentaceratops sternbergii, aff. Pentaceratops n. sp., Navajoceratops, and Terminocavus by the following characters: Median bar bears extensive lateral flanges extending into the parietal fenestrae. Flanges are extensive such that the cross section of the median bar is a broad flat lenticular shape, rather than being narrow and strap-like.

Description

Parietal - The preserved portion measures 31 cm in length and represents most of the parietal median bar (Fig. 6). As with many vertebrate fossils from the De-na-zin Member, NMMNH P-33906 has a thin covering of pale-colored concretion, and many adhered patches of hematite. This obscures fine surface details, although most morphological features can be discerned. The dorsal side is gently curved laterally, but otherwise has no obvious surface features (i.e., it lacks a prominent medial crest, ridge, or bumps). In contrast, the ventral side bears a raised central bar with lateral flanges that extend laterally into the fenestrae. The lateral flanges are much more strongly developed than in Pentaceratops, Navajoceratops, and Terminocavus, but overall the median bar is less broad than in Anchiceratops (with the possible exception of referred specimen CMN 8535; Sternberg, 1929; Mallon et al., 2011). The cross section is different at either end of the median bar, which is used to infer orientation. At the inferred anterior end, the cross section is concave-convex, with a shallowly concave ventral side. At the inferred posterior end, the cross section is biconvex and lenticular in shape. In other chasmosaurines the anterior end of the parietal median bar can be slightly concave ventrally (e.g., aff. Pentaceratops n. sp., MNA Pl. 1747; Rowe, Colbert & Nations, 1981; Chasmosaurus belli holotype CMN 491; Hatcher, Marsh & Lull, 1907), so we have identified the ventrally concave end as anterior in NMMNH P-33906. The median bar is expanded laterally at both ends; this is typical of chasmosaurine median bars, but is important as it helps constrain the size of the fenestrae. Lateral expansion is more notable at the posterior end, although this is probably due to the anterior end being less complete. At its narrowest point, the median bar is 9 cm wide.

Figure 6 Chasmosaurinae sp. “Taxon C” NMMNH P-33906 parietal median bar.

Near-complete parietal median bar in right lateral (A), dorsal (B), left lateral (C), ventral (D), and ventral outline (E) views. Cross sections in posterior (F) and anterior (G) inferred views. Subtle lateral expansion at both anterior and posterior ends suggests that the length of the median bar is complete, and as such is much wider than in stratigraphically preceding forms Utahceratops, Pentaceratops, Navajoceratops, and Terminocavus. The extra width is due to more extensive tapering lateral edges (te) of the median bar which extend out into the parietal fenestrae. Scalebar = 10 cm.

Epijugal - NMMNH P-33906 includes an epijugal which is fused to the jugal (and probably the quadratojugal). However, the jugal and quadratojugal are almost entirely missing, with the only remaining parts being small pieces that are fused to the base of the epijugal. The epijugal measures ~10 cm long, and is moderately pointed in shape.

Ontogenetic assessment

Significant morphologic change through ontogeny can strongly affect the phylogenetic placement of a specimen (Campione et al., 2013; Currie et al., 2016). It is therefore important to determine the ontogenetic status of new specimens so that appropriate comparisons can be made. No limb bones are preserved with the new specimens described here, so the age in years of individuals cannot be determined. Ontogenetic change in cranial morphology is not well studied in non-triceratopsin chasmosaurines (although see Lehman, 1990; Campbell et al., 2016), although it has been intensively studied in the derived chasmosaurine Triceratops (Horner & Goodwin, 2006, 2008; Scannella & Horner, 2010, 2011; Farke, 2011; Horner & Lamm, 2011; Longrich & Field, 2012; Maiorino et al., 2013). Based on this prior work, a combination of ontogenetically variable cranial features (size, sutural fusion, shape and fusion of epiossifications, frill surface texture, squamosal elongation) are here hypothesized to also be indicative of subadult or adult status in SMP VP-1500, NMMNH P-27468, and NMMNH P-33906.

Size - Size is an unreliable measure of maturity, as individual body size variation has been shown to be considerable in some dinosaurs (Sander & Klein, 2005; Woodward et al., 2015). Nevertheless, large size is often used as a rough gauge of maturity (and conversely, small size of immaturity), and this is a reasonable approach when used in combination with other morphological features that are ontogenetically informative. The holotype parietal of Navajoceratops, SMP VP-1500, is of comparable size to other specimens of Pentaceratops and related chasmosaurines (Fig. 7). The holotype parietal of Terminocavus, NMMNH P-27468, was described as small in the abstract by Sealey, Smith & Williamson (2005), but it is only slightly smaller than specimens of Pentaceratops (Fig. 7). The squamosal of NMMNH P-27468 has a reconstructed length of 94 cm, which is slightly smaller than MNA Pl.1747 (127 cm; J. Fry, 2015, personal communication), but larger than the juvenile aff. Pentaceratops SDMNH 43470 (77 cm; Diem & Archibald, 2005); the only other complete Pentaceratops squamosal is AMNH 1624, which is undescribed. The jugal of NMMNH P-27468 is only slightly smaller than Utahceratops referred specimen UMNH VP-12198 (Fig. S9), which is a large and aged individual (fused frill epiossifications that are mediolaterally elongate, spindle-shaped, and blunt; resorbed postorbital horns; fused epijugal; Sampson et al., 2010; D. Fowler, 2013, personal observations). The median bar of NMMNH P-33906 (Taxon C) is much broader than the median bar of any specimen of Pentaceratops, Navajoceratops, or Utahceratops (Fig. 7). At 10 cm long, the epijugal of NMMNH P-33906 is also of similar size to the epijugal of UMNH VP-12198.

Figure 7 Parietal relative sizes among specimens of Pentaceratops, and related chasmosaurines.

Parietals of chasmosaurine taxa mentioned in the main text, all in dorsal view and to scale with each other to show relative size. Taxa shown in stratigraphic order (with the exception of (E), SDMNH 43470). (A) Utahceratops gettyi referred specimen UMNH VP-16671; (B) cf. Pentaceratops sternbergii referred specimen AMNH 1625. Aff. Pentaceratops sp. referred specimens (C) UKVP 16100; (D) NMMNH P-37880, and (F) MNA Pl. 1747; (E) Aff. Pentaceratops sternbergii referred specimen SDMNH 43470; (G) Navajoceratops sullivani holotype SMP VP-1500; (H) Terminocavus sealeyi holotype NMMNH P-27468. (I) Chasmosaurinae sp. “Taxon C” specimen NMMNH P-33906. ep, epiparietal loci numbered by hypothesized position (no epiossifications are fused to this specimen); mb, median bar. Line drawings adapted from Longrich (2014), and Sampson et al. (2010). Scalebar = 10 cm.

Cranial fusion - Fusion of cranial sutures is often used as an indicator of maturity, but this is fraught with problems as the timing of suture closure may not be consistent between taxa (for example, the nasals and epinasal fuse relatively early in young subadult specimens of Triceratops horridus, whereas the congeneric T. prorsus these elements fuse in late subadulthood, to adulthood; Horner & Goodwin, 2006, 2008; Scannella et al., 2014). However, similar to size, degree of cranial fusion can be informative when used in conjunction with other data. Fusion of the epijugal to the jugal and quadratojugal is observed in all three of the new specimens (albeit based only a tentative identification in SMP VP-1500). In Triceratops, fusion of the epijugal to the jugal and quadratojugal occurs relatively late in ontogeny, as a subadult or adult (Horner & Goodwin, 2008). A similar survey has not been conducted for more basal chasmosaurines (but see Campbell et al., 2016), although the small-sized purportedly immature aff. Pentaceratops specimen SDMNH 43470 (Diem & Archibald, 2005) includes an unfused jugal and quadratojugal, but no epijugal as it was unfused and not recovered with the rest of the skull. Larger specimens of Pentaceratops and related taxa exhibit fusion of the epijugal to the jugal (holotype AMNH 6325, AMNH 1625, KUVP 16100; J. Fry, 2015, personal communication). From this, fusion of the epijugal in NMMNH P-27468 and P-33906 (also, tentatively SMP VP-1500; Figs. S7 and S9) is considered supportive of subadult or adult status.

Frill epiossifications - Shape and fusion of frill epiossifications varies through ontogeny in chasmosaurines. In Triceratops, the episquamosals fuse first, followed by the epiparietals (Horner & Goodwin, 2008).

Godfrey & Holmes (1995) suggest that in Chasmosaurus, fusion of the episquamosals begins at the anterior end of the squamosal, and proceeds posteriorly through ontogeny. This pattern is similarly observed in Pentaceratops and related taxa, notably in aff. Pentaceratops n. sp. MNA Pl. 1747 (Rowe, Colbert & Nations, 1981) and aff. P. sternbergii SDMNH 43470 (Diem & Archibald, 2005) in which only the anterior episquamosals are fused. Fusion of episquamosals in SMP VP-1500 (probably from the middle of the squamosal; Fig. S6) supports the identification of this specimen as a subadult or adult. NMMNH P-27468 only preserves the anteriormost fused episquamosal (the rest of the squamosal lateral border is damaged; Fig. S8), so it is consistent with subadult or adult status, but this cannot be confirmed without additional material or data on the timing of the fusion of the first episquamosal.

The order of epiparietal fusion is not studied in basal chasmosaurines (but see Campbell et al., 2016) and a specific pattern has not yet been identified for Triceratops. However, a survey of specimens referred to Pentaceratops (and related taxa) reveals a general pattern where ep1 fuses first, followed by ep2, then ep3. Ep1 is fused in the four largest specimens (cf. P. sternbergii AMNH 1625, aff. Pentaceratops n. sp. MNA Pl. 1747, KUVP 16100, and cf. Utahceratops UMNH VP-16671 and 16784; Fig. 7), but is unfused in the aff. P. sternbergii small specimen SDMNH 43470, and in newly described parietal fragment NMMNH P-37890 (see Supporting Information 1). Ep2 is fused in AMNH 1625, MNA Pl. 1747, UMNH VP-16671 and 16784, but not in KUVP 16100. Ep3 is fused in AMNH 1625, UMNH VP VP-16671 and 16784, and possibly MNA Pl. 1747 (see Supporting Information 1), but is unfused in KUVP 16100. The Navajoceratops holotype SMP VP-1500 has fused ep1 (probable) and ep2, but ep3 is unfused hence it exhibits a state of fusion between KUVP 16100 and MNA Pl.1747 (or AMNH 1625), and on this basis could be considered subadult. The holotype of Terminocavus (NMMNH P-27468) has fused ep1 and ep2 on both sides; ep3 is fused only on the left side, with an open space on the right side at the ep3 locus. On this basis, NMMNH P-27468 should be considered subadult or adult.

Regarding shape, all Triceratops frill epiossifications develop from being triangular-shaped with pointed apices and short bases in juveniles, to spindle shaped with blunt apices and elongate bases in adults (Horner & Goodwin, 2006, 2008). Similar patterns exist in the episquamosals of more basal chasmosaurines with probable juvenile and immature specimens of Chasmosaurus (Campbell et al., 2016), Agujaceratops (Lehman, 1989) aff. Pentaceratops (SDMNH 43470; Diem & Archibald, 2005), and Arrhinoceratops (Mallon et al., 2014) exhibiting more short-based, pointed episquamosals. The episquamosals of Navajoceratops holotype SMP VP-1500 (Fig. S6) are spindle shaped, and blunt with elongate bases, consistent with a subadult or adult condition. The Terminocavus holotype, NMMNH P-27468, only has the anteriormost episquamosal preserved, which tends to remain triangular and slightly pointed in subadult and adult chasmosaurines, even when more posterior episquamosals develop into spindle shapes. Thus, the triangular shape of the episquamosal of NMMNH P-27468 is not ontogenetically informative. Note that triceratopsins are slightly unusual among chasmosaurines in that their epiparietals and episquamosals are of similar morphology to each other; whereas in Anchiceratops and more basal chasmosaurines, the epiparietals take a greater variety of forms. Most notable is that the epiparietals remain large and triangular through to adulthood in Utahceratops, cf. and aff. Pentaceratops, Navajoceratops, and particularly Terminocavus and Anchiceratops.

Frill surface texture - The texture of the parietosquamosal frill (and many of the facial bones) has been shown to change ontogenetically in both centrosaurine and chasmosaurine ceratopsids (Sampson, Ryan & Tanke, 1997; Brown, Russell & Ryan, 2009; Scannella & Horner, 2010). Adult ceratopsids are characterized by a distinctive frill texture where indented vascular channels form complex dendritic patterns. This texture gradually develops through ontogeny, with juveniles exhibiting a smooth or “long-grain” bone texture (Sampson, Ryan & Tanke, 1997; Brown, Russell & Ryan, 2009; Scannella & Horner, 2010), which is replaced by a pebbled or pitted texture with shallowly developed vascular traces in young subadults. This is complicated somewhat by recognition that this long-grain texture is associated with rapid growth (Francillon-Vieillot et al., 1990; Sampson, Ryan & Tanke, 1997) and/or expansion of the frill, as expected in juveniles, but is also seen in some specimens of Torosaurus which were reshaping their frills relatively late in ontogeny (Scannella & Horner, 2010). The Navajoceratops holotype SMP VP-1500 has well developed adult frill texture on both the parietal (Fig. 4) and the squamosal (Fig. S6). In the Terminocavus holotype, NMMNH P-27468, the frill texture on the parietal is partially obscured by a thin layer of sediment covering the surface, but can be seen to be pitted with shallow vascular canals. The same texture is visible on the dorsal surface of the squamosal. This suggests that NMMNH P-27468 was not yet fully mature and may be considered a young subadult. Surface texture is not discernible on Taxon C specimen NMMNH P-33906.

Squamosal elongation - In juvenile chasmosaurines, the squamosal is anteroposteriorly short, similar to the condition in adult centrosaurine ceratopsids and more basal neoceratopsians (Lehman, 1990; Goodwin et al., 2006; Horner & Goodwin, 2006; Scannella & Horner, 2010; Mallon et al., 2014; Campbell et al., 2016). In chasmosaurines, the squamosal elongates through ontogeny, although the timing of the elongation varies phylogenetically (Lehman, 1990; Scannella & Horner, 2010). The derived taxon Triceratops has been shown to retain an anteroposteriorly short squamosal until relatively late in ontogeny (Scannella & Horner, 2010), whereas in Chasmosaurus and Pentaceratops (albeit based on more limited data) it would appear that elongation occurs at smaller body sizes (inferred to be younger; Lehman, 1990). Although the squamosal of SMP VP-1500 comprises only fragments, one fragment (Figs. S6C and S6D) might represent the more bladed posterior end, which would be supportive of a subadult or adult status. The squamosal of NMMNH P-24768 is incomplete, but enough remains to show that it was relatively elongate, supporting a subadult or adult status.

Geometric morphometric analysis

Results of the geometric morphometric Principal Components Analysis (PCA) on chasmosaurine parietals are presented in Fig. 8. PC 1 (x-axis) accounts for 50.5% of variation, and assesses depth of the median embayment from shallow (negative) to deep (positive), and orientation of the base of ep1 from mediolateral (negative) to anteroposterior (positive); PC 2 (y-axis) accounts for 19.0% of variation and assesses lateral expansion of the ep1 locus, shape of the posterolateral corner of the parietal, and overall anteroposterior length.

Figure 8 Morphometric analysis of chasmosaurine posterior parietals.

Deformation grids illustrate shape of left lateral ramus and median bar of each specimen at the end of each principal component axis (PC). Colored dots on deformation grids represent landmarks illustrated in Fig. 3. PC 1 (x axis) accounts for 50.5% of variation and assesses depth of the median embayment from shallow (negative) to deep (positive), and orientation of base of ep1 from mediolateral (negative) to anteroposterior (positive). PC 2 (y axis) accounts for 19.0% of variation. Points connected by a bar represent left and right sides of the same specimen (where adequately preserved). Pentaceratops through Anchiceratops plot along PC 1, demonstrating progressively deeper median embayment, and an increase in the angle of ep1. Chasmosaurus through to Vagaceratops are concentrated on the negative side of PC 1, following a trend from positive to negative along PC 2. Key: “Ag”, Agujaceratops; An, Anchiceratops; Ch.b, Chasmosaurus belli; cf. Ch.r, cf. "Chasmosaurus russelli"; Ko, Kosmoceratops; Na, Navajoceratops; aff. Pe n.sp., aff. Pentaceratops n. sp.; cf. Pe, cf. Pentaceratops sternbergii; Te, Terminocavus; Ut, Utahceratops; Va, Vagaceratops. Color to aid in distinction only.

Specimens previously assigned to the same taxon largely cluster into groups, with “Chasmosaurus russelli”, C. belli, and Anchiceratops specimens all clustering together. Specimens referred to cf. Pentaceratops n. sp (MNA Pl.1747 and KUVP 16100) are separated from cf. P. sternbergii specimen AMNH 1625, justifying their consideration as different taxa. The new taxa, Navajoceratops and Terminocavus, plot as intermediate between these stratigraphically preceding chasmosaurines and the stratigraphically higher Anchiceratops.

Two perpendicular morphological trends correlate with the stratigraphic occurrence of taxa and match the lineages proposed by Lehman (1998). From stratigraphically oldest to youngest, “Chasmosaurus russelli”, C. belli, and Vagaceratops irvinensis occupy the negative end of the PC 1 axis, and are spread down the PC 2 axis in stratigraphic order, showing little variation along the PC 1 axis. This demonstrates progressive expansion of the ep1 locus, concentrating ep2 and ep3 to the lateralmost corner of the parietal. The trend in Chasmosaurus is contrasted by a second group (comprising Utahceratops, Pentaceratops, Navajoceratops, Terminocavus, and Anchiceratops) which is mostly distributed along the PC 1 axis in stratigraphic order, and shows relatively little variation on PC 2. This group exhibit progressive deepening and eventual closure of the median embayment, an increasingly steep angle of the ep1 locus, and anteroposterior expansion of the posterior bar.

There are some inconsistencies in that Kosmoceratops does not plot close to Vagaceratops on the PC 1 axis (although it is very close on the PC 2 axis), despite being recovered as sister taxa in most phylogenetic analyses (Sampson et al., 2010; Mallon et al., 2014; and this analysis, see below). Similarly, aff. Pentaceratops n. sp. specimen MNA Pl.1747 plots more negatively on the PC 2 axis than other specimens within the Pentaceratops grouping (although it is very similarly placed along the PC 1 axis). These issues might be a reflection of potential problems with the input data concerning these two specimens. First, for Kosmoceratops, points were plotted on to the dorsal view provided by Sampson et al. (2010). However, this is not completely perpendicular to the parietal surface. Consultation of photographs of skull casts shows that the parietal posterior bar of Kosmoceratops is not as medially embayed as it appears in the image used (this being an artifact of slight arching of the parietal). Hence it is predicted that upon reanalysis of a perpendicular photograph, Kosmoceratops might plot more negative along PC 1 (x axis), closer to other members of the Chasmosaurus clade. Second, aff. Pentaceratops n. sp. MNA Pl.1747 may require revision if the redescription of J. Fry indeed identifies that ep3 is fused to the posterolateral corners of the parietal. This would reduce the anteroposterior offset of the lateralmost margin of the parietal, bringing the morphology of MNA Pl.1747 more similar to KUVP 16100.

Phylogenetic analysis

Phylogenetic analysis recovers Navajoceratops sullivani and Terminocavus sealeyi as close relatives of both Pentaceratops and Anchiceratops. The initial analysis was run using the amended matrix of Mallon et al., 2014 (see Supporting Information 2), with only Mojoceratops perifania excluded because this is considered a junior synonym of Chasmosaurus russelli (Maidment & Barrett, 2011; Mallon et al., 2011). This resulted in 6 most parsimonious trees (L = 319 steps; CI = 0.72; RI = 0.79). The strict consensus tree (Fig. 9A) supports a monophyletic Chasmosaurinae, and recovered Navajoceratops and Terminocavus as successive sister taxa to Anchiceratops, Arrhinoceratops, and Triceratopsini. However, (Pentaceratops + Utahceratops) + (Coahuilaceratops + Bravoceratops) is recovered as sister group to this clade, rather than a direct relationship between Pentaceratops and Navajoceratops, as would have been predicted based on parietal morphology. A basal Chasmosaurus clade was separated from a (Vagaceratops + Kosmoceratops) clade by Agujaceratops.

Figure 9 Phylogenetic analysis.

(A) Strict consensus tree showing all taxa (MPT = 6; L = 319; CI = 0.72; RI = 0.79). (B) Reanalysis 1, strict consensus tree (MPT = 6; L = 310; CI = 0.72; RI = 0.79). Bravoceratops, Agujaceratops removed from the character matrix. (C) Strict consensus tree showing all taxa (MPT = 28; L = 308; CI = 0.72; RI = 0.79). Bravoceratops, Agujaceratops, Coahuilaceratops removed from the character matrix Numbers on nodes indicate bootstrap values >50%; nodes without values had <50% support. Character matrix altered from Sampson et al. (2010) and Mallon et al. (2014).

Reanalysis 1 additionally excluded nomen dubium Bravoceratops, and Agujaceratops because it is coded partly from juvenile material and specimens that may not be referred to the taxon (see Supporting Information 1). This yielded six most parsimonious trees (Length (L) = 310 steps; Consistency Index (CI) = 0.72; Retention Index (RI) = 0.79). The strict consensus tree (Fig. 9B) maintains the relationship of [Utahceratops + Pentaceratops + Coahuilaceratops] as sister group to (Navajoceratops + Terminocavus + Anchiceratops + Arrhinoceratops + Triceratopsini). The most significant result of reanalysis 1 is the unification of a Chasmosaurus clade with (Vagaceratops + Kosmoceratops). This is similar to the original description of Vagaceratops (Chasmosaurus) irvinensis (Holmes et al., 2001), where the taxon was considered the most derived (and stratigraphically youngest) form of Chasmosaurus, a relationship also recovered in the phylogenetic analyses of Longrich (2014) and Campbell et al. (2016, 2019).

Reanalysis 2 investigated the effect of excluding Coahuilaceratops from the dataset because Coahuilaceratops is known from very fragmentary material. This yielded 28 most parsimonious trees (L = 308; CI = 0.72; RI = 0.79). The strict consensus tree (Fig. 9C) maintained the basal Chasmosaurus clade, but Utahceratops, Pentaceratops, Navajoceratops, Terminocavus, and Anchiceratops collapsed into a polytomy.

These analyses support the finding of the morphometric analysis in that the new taxa Navajoceratops and Terminocavus are morphological intermediates between Pentaceratops and Anchiceratops, although the absence of a sister group relationship between Navajoceratops and Pentaceratops is not supportive of evolution by anagenesis. However, this may be due to the way that P. sternbergii is coded in this dataset (see below). The topologies of reanalysis 1 and 2 also supports the proposal of Lehman (1998) that a deep split divides the Chasmosaurinae into two lineages.

These results are closer to matching the evolutionary hypotheses based on the stratigraphic positions of taxa, but represent only a first step in the many revisions required of the phylogenetic matrix. Most significant to this study is that in the current matrix, the composite coding of P. sternbergii includes specimens that are probably not all referable to the same taxon, for example, AMNH 6325, 1624, 1625, NMMNH P-50000, and those considered here as aff. Pentaceratops n. sp. (MNA Pl.1747 and KUVP 16100). It is therefore required for these specimens to be coded and analysed as at least three separate taxa, but this action awaits the description of the anterior skull elements of these specimens currently being completed by Joshua Fry. A similar recoding is required for Agujaceratops; the immature holotype material should not be used for coding the taxon, as its immature status may affect its phylogenetic positioning (Campione et al., 2013). Instead, referred specimens UTEP P.37.7.065 (isolated parietal) and TMM 43098-1 (near-complete skull, missing the parietal) should be coded separately. The holotype of Chasmosaurus russelli (CMN 8800) requires redescription, and will likely need to be moved out of Chasmosaurus and coded separately from other referred specimens (Longrich, 2015; Campbell et al., 2016; Fowler & Freedman Fowler, 2017). Chasmosaurus belli referred specimen YPM 2016 has been redescribed (Campbell et al., 2019), and will need to be coded separately into our new matrix as a morphologic intermediate between C. belli specimens and Vagaceratops. Finally, some recently described chasmosaurine taxa (e.g., Judiceratops; Mercuriceratops; Regaliceratops, and Spiclypeus; Longrich, 2013; Ryan et al., 2014; Brown & Henderson, 2015; Campbell, 2015; Mallon et al., 2016) have yet to be coded into the revised matrix, although new taxa known from fragmentary remains may require some reassessment which is beyond the scope of this current work.

Discussion

Comparisons and discussion of morphological characters

As the holotype specimens are probable subadults or adults, Navajoceratops and Terminocavus can be appropriately compared with other taxa that are based on putative adults.

Navajoceratops and Terminocavus form progressive morphological intermediates between the stratigraphically preceding Pentaceratops and succeeding Anchiceratops. Although limited in available material, “Taxon C” (NMMNH P-33906) exhibits morphology intermediate between the stratigraphically preceding Terminocavus, and succeeding Anchiceratops. A number of characters of the parietal provide the best means to compare among chasmosaurine taxa.

Before continuing, it should be noted that the evolutionary patterns suggested here for the Chasmosaurus clade (Chasmosaurus + Vagaceratops + Kosmoceratops) represent one interpretation of the available data. An alternative hypothesis is offered by Campbell et al. (2016, 2019). Differences in opinion center around interpretation of specific Chasmosaurus specimens, notably the stratigraphic position of YPM 2016, the comparability of CMN 2245 based on its ontogenetic status, the taxonomic affinity of fragmentary specimens, and the nature of variation in a population (see Supporting Information 1). Although we prefer the explanation of an anagenetic evolutionary mode for both Chasmosaurus and Pentaceratops lineages, Campbell et al. (2019) suggest a cladogenetic origin for at least C. sp. The morphologic trends described here are consistent with the hypothesis that Kosmoceratops evolved from Vagaceratops, which evolved from C. sp, C. belli, and "C. russelli", regardless of evolutionary mode.

Median embayment of the posterior bar

The median embayment of the posterior bar is one of the most important morphological features in distinguishing chasmosaurine taxa. It is defined by the angle at which the lateral rami meet medially, and the proportion of the posterior bar occupied by the embayment.

The angle at which the lateral rami of the posterior bar meet medially (see Figs. S10 and S11) is similar in more basal chasmosaurines, but becomes disparate in more derived forms. Within chasmosaurines allied to Chasmosaurus, the lateral rami meet at a relatively shallow angle, measuring 87–131° in specimens referred to “C. russelli”, and is shallower in stratigraphically overlying taxa C. belli/C. sp.(149–167°) and Vagaceratops (177°). In contrast, the lateral rami meet at a relatively steep angle in Utahceratops (75°), cf. Pentaceratops sternbergii (83°), and aff. Pentaceratops n. sp. (87–88°). Navajoceratops (60°) and Terminocavus (~73°) exhibit angles that are more acute than stratigraphically preceding chasmosaurines, indicating the deepening and enclosing of the median embayment. However, in Terminocavus and especially Anchiceratops, measurement of the angle of the lateral rami is not straightforward as the lateral rami have become curved and anteroposteriorly expanded (although in theory, closure of the deep embayment in Anchiceratops means that the angle could be considered as 0°).

The median embayment is restricted to the central 30–50% of the posterior bar in stratigraphically older chasmosaurines such as “Chasmosaurus russelli”, Agujaceratops, Utahceratops, and cf. Pentaceratops sternbergii. In more derived forms, the apex of the arch formed by each lateral bar migrates towards the lateral margin, broadening the median embayment. In C. belli, C. sp, Vagaceratops and (to an extent) Kosmoceratops, this occurs concomitantly with an increase in the angle of the lateral bars such that the embayment appears weakened or lost. In contrast, in aff. Pentaceratops sp., the angle increases, and the embayment appears deeper. In Navajoceratops and Terminocavus the embayment is again restricted to the central 30–50% of the posterior bar, mainly because anteroposterior expansion of the posterior bar at the ep3 locus gives the lateral bars a more rounded shape. In Anchiceratops, the median embayment is effectively completely closed, with only a shallow depression remaining between left and right ep2.

Epiparietal Number, shape, size, and orientation

Chasmosaurines typically exhibit three epiparietal loci on each side. Important morphological differences among taxa include shape and size of all epiparietals; position and consequent orientation of ep1 and ep2 relative to the median embayment of the parietal posterior bar; position and orientation of ep3 relative to the posteriormost point of the posterior bar and the articulation with the squamosal.

Of the new specimens, ep1 is only preserved in Terminocavus holotype NMMNH P-27468, where its triangular shape is comparable to cf. Pentaceratops sternbergii, aff. Pentaceratops n. sp., Anchiceratops, and some specimens referred to "Chasmosaurus russelli", and unlike the laterally expanded ep1 locus in C. belli, C. sp. Vagaceratops, and Kosmoceratops. In Terminocavus ep1 is only slightly deflected dorsally, comparable to the right side of cf. P. sternbergii AMNH 1625, and "P. aquilonius" referred specimen CMN 9814 (Longrich, 2014), rather than folded over the posterior bar to point anterolaterally (as in the left side of cf. P. sternbergii AMNH 1625, and aff. Pentaceratops n. sp.) or laterally (Anchiceratops). Given its phylogenetic position, it might be expected for Terminocavus to exhibit an anterolaterally oriented ep1 rather than being only slightly deflected dorsally. It is possible that ep1 folds over anteriorly through ontogeny, and that the condition in NMMNH P-27468 is indicative that it is not fully mature; ontogenetic indicators (see above) suggest a status between young subadult to adult for NMMNH P-27468, which leaves open the possibility that the epiparietals might have folded anteriorly if the individual had survived to later greater maturity. However, different ep1 orientations between left and right sides of the putative adult cf. P. sternbergii, AMNH 1625, demonstrates that this character is variable, even in an adult.

In Navajoceratops and Terminocavus locus ep1 occurs within the median embayment, as in Utahceratops, cf. Pentaceratops sternbergii and aff. Pentaceratops n. sp. This is unlike cf. Agujaceratops (UTEP P.37.7.065) and specimens referred to “Chasmosaurus russelli” where ep1 occurs at the edge of the embayment. In C. belli, C. sp. Vagaceratops, and Kosmoceratops, the ep1 locus is expanded laterally and occupies most of the posterior bar (see Supporting Information 1 and Fig. S18). In contrast, in Anchiceratops, the median embayment is closed such that ep1 effectively occurs at the midline on the dorsal surface of the posterior bar. Orientation of the long axis of ep1 follows the angle of the lateral rami upon which it is mounted. In Chasmosaurus it is therefore oriented mostly mediolaterally. In contrast, ep1 is oriented slightly anteroposteriorly in cf. Pentaceratops sternbergii, and at an increasingly steep angle from cf. P. sternbergii through Navajoceratops, Terminocavus, and finally Anchiceratops in which it is oriented anteroposteriorly such that the tips point laterally.

In both Navajoceratops and Terminocavus holotypes ep2 is large and triangular; in Navajoceratops the apices are broadly rounded apices rather than being pointed, whereas in the Terminocavus holotype, both ep2 have damaged apices. Large triangular ep2 are seen in most chasmosaurines, although these reach especially large size in Anchiceratops. Ep2 is cryptic in specimens of C. belli and C. sp., with its identification depending on the epiossification numbering scheme and opinion of the researcher (see Supporting Information 1 and Fig. S18). Ep2 is anteriorly inclined in Vagaceratops, and Kosmoceratops. In the derived Triceratops all frill epiossifications are triangular in juveniles, and become broad and flattened in adults (Horner & Goodwin, 2006).

In Navajoceratops, ep2 occurs within the median embayment and the pointed tip is medioposteriorly oriented, as in aff. Pentaceratops n. sp., and unlike the stratigraphically preceding cf. P. sternbergii and Utahceratops, where ep2 points posteriorly. In Terminocavus, the position and orientation of ep2 is intermediate between Navajoceratops and Anchiceratops; anteroposterior expansion and increased curvature of the lateral rami causes the constriction of the median embayment such that ep2 is less medially oriented than in Navajoceratops, and closer to a posterior orientation.

Locus ep2 is the posteriormost locus in basal chasmosaurines “Chasmosaurus russelli”, most specimens of C. belli, C. sp., Kosmoceratops, Utahceratops, and cf. Pentaceratops sternbergii. The posteriormost epiparietal locus switches to ep3 in chasmosaurines more derived than cf. P. sternbergii (aff. Pentaceratops n. sp, Navajoceratops, Terminocavus, and Anchiceratops).

In chasmosaurines, the apex of locus ep3 points laterally or posterolaterally in “Chasmosaurus russelli”, posterolaterally in C. belli; C. sp. Vagaceratops, Kosmoceratops, Utahceratops, and cf. Pentaceratops sternbergii; and posteriorly in aff. Pentaceratops n. sp. (inferred from locus), Navajoceratops (inferred from locus), Terminocavus, and Anchiceratops.

Anteroposterior thickness of the posterior bar lateral rami

The anteroposterior thickness of the posterior bar is narrow and strap-like in more basal chasmosaurines (Chasmosaurus, Vagaceratops, Kosmoceratops, Utahceratops, cf. Pentaceratops sternbergii), broadening to become flat and plate-like in the most derived forms (Anchiceratops, Arrhinoceratops, and Triceratopsini). In Navajoceratops the posterior bar is anteroposteriorly expanded laterally, being broadest at locus ep3. This is also exhibited by the stratigraphically preceding aff. Pentaceratops n. sp., but is unlike cf. Pentaceratops sternbergii, Utahceratops, Chasmosaurus, and Vagaceratops, where the posterior bar is strap-like and subequal in anteroposterior thickness along its length. In Terminocavus the lateral rami are much more similar to Anchiceratops in being strongly anteroposteriorly expanded such that they are plate-like rather than bar-like.

Characters of the median bar and parietal fenestrae

The parietal median bar exhibits two characters that differ among taxa; the anteroposterior position of the point of maximum constriction, and the development of lateral flanges which invade the parietal fenestrae (with consequent effect on the shape of the median bar cross section).

In referred specimens of “Chasmosaurus russelli”, C. belli, C. sp.and Kosmoceratops, the point of maximum constriction occurs in the posteriormost third of the median bar. In most specimens of C. belli, this is immediately at the point of contact with the posterior bar. In Vagaceratops irvinensis, the median bar is slightly damaged, but the preserved portion also seems to have the point of maximum constriction in the distal third. In contrast, in cf. Pentaceratops sternbergii, aff. P. n. sp., Anchiceratops, Arrhinoceratops, and fenestrated Triceratopsini, the point of maximum constriction occurs approximately at the anteroposterior midpoint of the median bar. The median bar is incomplete in parietals of cf. Agujaceratops, Utahceratops, Navajoceratops, Terminocavus, and Chasmosaurinae sp. “taxon C” (NMMNH P-33906), but in these taxa the maximum constriction does not occur adjacent to the posterior bar (i.e., as in Chasmosaurus), and probably occurs approximately half way along its length.

In basal chasmosaurines Chasmosaurus, Agujaceratops, Utahceratops, cf. Pentaceratops sternbergii, aff. Pentaceratops n. sp., and Navajoceratops the median bar is narrow and strap-like, but develops into a broader structure in Vagaceratops (slightly), Kosmoceratops, and especially from Terminocavus through Chasmosaurinae sp. “taxon C”, Anchiceratops, Arrhinoceratops, and Triceratopsini. Broadening of the median bar is therefore possibly convergent between Chasmosaurus and Anchiceratops clades. In the taxa basal to Anchiceratops, broadening occurs by development of thin lateral flanges which project from the lateral edges of the median bar, generally only easily observable on the ventral side. These are very weakly developed in Utahceratops referred specimen UMNH VP-16671, and remain weak to absent in cf. P. sternbergii and aff. Pentaceratops n. sp. In Navajoceratops they are slightly more prominent than in stratigraphically preceding taxa, and are similarly further developed in Terminocavus. Lateral flanges are much more developed in the stratigraphically younger “taxon C” (NMMNH P-33906; Fig. 6), where they approach the level of development seen in some specimens of Anchiceratops (e.g., CMN 8535; TMP 1983.001.0001; Mallon et al., 2011). Development of lateral flanges is associated with the reduction in size, and change in shape of the parietal fenestrae.

An obvious character that differentiates basal and derived chasmosaurines is the size and shape of the parietal fenestrae. The fenestrae of derived chasmosaurines (Kosmoceratops, Anchiceratops, Arrhinoceratops, and Triceratopsini) are subrounded to subcircular (although only subangular to subrounded in Kosmoceratops), relatively small, and enclosed within the parietal by a broad median bar and wide parietal lateral bars. This is contrasted with the large angular to subangular fenestrae of basal chasmosaurines (Chasmosaurus, Vagaceratops, Utahceratops, cf. Pentaceratops sternbergii, aff. P. n. sp., and Navajoceratops) which are typically enclosed only by a narrow median bar and thin lateral bars which may not be anteroposteriorly continuous (hence part of the squamosal may form the lateral border of the fenestra). Terminocavus is morphologically and stratigraphically intermediate between the two morphotypes, and has subrounded parietal fenestrae. Because “taxon C” is incomplete it is not possible to know the shape of the fenestrae.

The parietal fenestrae of ceratopsian dinosaurs open and expand in size through ontogeny (Dodson & Currie, 1988; Brown, Russell & Ryan, 2009; Scannella & Horner, 2010; Fastovsky et al., 2011; Currie et al., 2016). As such, it is possible that smaller and more rounded parietal fenestrae in Terminocavus holotype NMMNH P-27468 may indicate that the individual was not fully mature, and that the fenestrae would have been larger and perhaps more angular in the final growth stage. Although this is possible, the purportedly juvenile aff. Pentaceratops sp. SDMNH 43470 has fenestrae that are relatively larger and more angular (inferrable from the strap-like and straight posterior bar) than in the Terminocavus holotype which ontogenetic indicators suggest is a subadult or adult. As such, it is hypothesized that the final size and shape of the fenestrae might not be significantly different from that observed.

Implications of findings

Although this study demonstrates that most chasmosaurine taxa are still in need of detailed revision, the description of the new taxa provides a good basis from which to investigate the paleobiology of Chasmosaurinae as a group, and the influence of these findings on our understanding of dinosaur evolution in the Late Cretaceous of North America.

Phylogeny: anagenetic stacks of stratigraphically segregated “species”?

In his discussion on the validity of the badly distorted “Pentaceratops fenestratus”, Mateer (1980; p. 52) suggested that “the presence of two species (of Pentaceratops) in the San Juan Basin separated stratigraphically may be real”. The new taxa Navajoceratops and Terminocavus, along with taxon C (NMMNH P-33906), corroborate this view with better preserved material, expanding it beyond only two taxa, and provide critical morphological links between the stratigraphically preceding form Pentaceratops and succeeding Anchiceratops.

It is important to recognize that there is little evidence that the naming of these new taxa represents increased diversity; rather, the new taxa support identification of an unbranching lineage linking Pentaceratops and Anchiceratops, consistent with the hypothesis of Lehman (1998). The term “diversity” is used broadly in paleontology, typically when referring to multiple named species within a given clade as evidence of diversity. This is often inappropriate; “diversity” should properly only be used to denote two or more contemporaneous species or lineages. In this usage, diversity is evidence of lineage splitting or multiplication, also termed cladogenesis (sensu Rensch, 1959) or speciation (sensu Cook, 1906; Vrba, 1985). The new taxa provide little evidence of lineage splitting, being instead more supportive of an unbranching lineage of stratigraphically separated taxa (“anagenesis”; Rensch, 1959, used here sensu Wiley, 1981; syn. “phyletic evolution”; Simpson, 1961) from Utahceratops through Pentaceratops, Navajoceratops, Terminocavus, and Anchiceratops. The morphometric analysis strongly supports this anagenetic lineage, with each taxon recovered progressively more positive along the PC1 axis (Fig. 8). The phylogenetic analysis is less supportive of such a long lineage, with (Utahceratops + Pentaceratops) forming a separate clade to (Navajoceratops + Terminocavus + Anchiceratops). However, it is expected that this might not be a problem when specimens of cf. Pentaceratops sternbergii (e.g., AMNH 1625), which show strong similarity with Utahceratops, are coded separately from aff. P. n. sp. (MNA Pl.1747; KUVP 16100). However, this awaits full description of the aff. P. n. sp. materials. Since each of the new taxa is stratigraphically separated from preceding and succeeding forms, and stratigraphically preceding forms are recovered as less derived, then we fail to reject the hypothesis that they are transitional forms within a single unbranching lineage (note that if Navajoceratops and Terminocavus represent intermediate forms within an anagenetic lineage then it is arguable that they should be considered as a single species, rather than new species or genera; see Supporting Information 1).

Phylogeny: a deep-split Chasmosaurinae

A deep split within a monophyletic Chasmosaurinae is suggested by the morphometric and phylogenetic analyses, supported by stratigraphic data, and consistent with the proposed lineages of Lehman (1998). The split divides Chasmosaurinae into two clades: a Chasmosaurus clade (“C. russelli” + C. belli + Vagaceratops + Kosmoceratops) and a Pentaceratops clade (Utahceratops + Pentaceratops + Navajoceratops + Terminocavus + Anchiceratops + Arrhinoceratops + Triceratopsini). With the exclusion of (Arrhinoceratops + Triceratopsini) (see later discussion) both clades comprise stratigraphically separated taxa which either do not overlap, or may overlap slightly (Fig. 10; see Supporting Information 1 and Campbell et al., 2019), with the oldest forms more basal, and younger forms more derived. This is supportive of an initial cladogenesis (speciation) event which created two resultant lineages or clades.

Figure 10 Stratigraphic positions of chasmosaurine taxa.

Morphospecies of Chasmosaurus (A–D) and Pentaceratops (E–J) clades which do not overlap stratigraphically. These are hypothesized to form two anagenetic lineages which resulted from a cladogenetic branching event prior to the middle Campanian. (A) “Chasmosaurus russelli”, lower Dinosaur Park Fm, ~76.8 Ma. (B) Chasmosaurus belli, middle Dinosaur Park Fm, ~76.5–76.3 Ma. (C) Vagaceratops irvinensis, upper Dinosaur Park Fm, ~76.1 Ma. (D) Kosmoceratops richardsoni, middle Kaiparowits Fm, ~76.0–75.9 Ma. (E) Utahceratops gettyi, middle Kaiparowits Fm, ~76.0–75.6 Ma. (F), c.f. Pentaceratops sternbergii, unknown occurrence within “Fruitland Formation” ~76.0–75.1 Ma. (G) Aff. Pentaceratops n. sp., uppermost Fossil Forest Mbr, Fruitland Fm, ~75.1 Ma. (H) Navajoceratops sullivani, lowermost Hunter Wash Mbr, Kirtland Fm, ~75.0 Ma. (I) Terminocavus sealeyi, middle Hunter Wash Mbr, Kirtland Fm, ~74.7 Ma. (F) Anchiceratops ornatus, Drumheller to Morrin Mbr, Horseshoe Canyon Fm, ~71.7–70.7 Ma. Stratigraphic positions and recalibrated radiometric dates from Supporting Information 1 and Fowler (Chapter 2). Timescale from Gradstein et al. (2012). Specimens not to scale. Images adapted from Lehman (1998); Holmes et al., 2001; Sampson et al. (2010); Maidment & Barrett (2011); and Longrich (2014).

The two clades are characterized by a number of divergent, often opposite, morphological trends (expanded from those proposed by Lehman (1998)). Basal members of both clades exhibit an anteroposteriorly narrow parietal posterior bar bearing a median embayment, and three discrete epiparietals. In the Chasmosaurus clade the median embayment shallows in more derived taxa as ep1 expands laterally, ep2 and ep3 loci migrate to the posterolateral corners of the parietal, the posterior bar remains anteroposteriorly narrow, and the apices of the curved lateral rami of the posterior bar migrate laterally but remain at ep1 or ep2. This is contrasted with the Pentaceratops clade where the median embayment deepens and closes in on itself, ep1 remains medial but rotates its long axis such that it becomes anteroposteriorly oriented, ep2 and ep3 become large and triangular (maintained in adults), and the posterior bar becomes anteroposteriorly broad and plate-like with rounded lateral rami, the apex of which occurs at locus ep3. Some morphologic trends are parallel between the clades. The parietal fenestrae of both clades exhibit a trend towards reduction in size, and increase in roundedness, concomitant with laterally expanded median and lateral bars.

The phylogenetic pattern, morphological trends, and stratigraphic occurrence imply divergence from a common ancestral population. The oldest known representative of either clade are specimens referred to “Chasmosaurus russelli” (not including the holotype; see Supporting Information 1) from the lower part of the Dinosaur Park Formation (Holmes et al., 2001; Mallon et al., 2012; see Supporting Information 1). This horizon is radiometrically dated as between 77 and 76.3 Ma, corresponding to the uppermost part of the middle Campanian (Eberth, 2005, 2011; Fowler, 2017). The oldest member of the Pentaceratops clade, Utahceratops, is slightly younger than this at between ~75.97 Ma to ~75.6 Ma (Roberts et al., 2013; Fowler, 2017). The cladogenetic split between Chasmosaurus and Pentaceratops clades must therefore have occurred before 77 Ma.

Collection of new chasmosaurine material from before 77 Ma is thus essential to further our understanding of the timing, rate, and cause of the divergence. Appropriately-aged dinosaur-bearing formations in the Western Interior include the Foremost (~80.2–79.4 Ma) and Oldman Formations, Alberta (~79.4–77Ma); lower parts of the Judith River (~80–77 Ma) and Two Medicine (~81–75 Ma) Formations, Montana; Wahweap Formation, Utah (~80 to ~79 Ma), and possibly the Aguja Formation, Texas (lower to middle Campanian; Goodwin & Deino, 1989; Rogers, Swisher & Horner, 1993; Rogers & Swisher, 1996; Jinnah, 2013; Roberts et al., 2013; Fowler, 2017; see Supporting Information 1). Although a good amount of material has been collected from the Aguja Formation (Lehman, 1989; Forster et al., 1993), most is fragmentary, immature, or is missing the critical parietal, making comparisons difficult. However, an isolated middle portion of the parietal posterior bar (UTEP P.37.7.065) is tantalizingly similar to basal members of both Chasmosaurus and Pentaceratops clades in exhibiting a median embayment restricted to the middle third, however, more complete parietal material is required for further comparisons (also see Supporting Information 1). A range of material has also recently been collected from the Judith River Formation of Montana and lower Oldman of southern Alberta (some published, e.g., the highly fragmentary remains named Judiceratops tigris; Longrich, 2013; Campbell, 2015) which has great potential to increase our knowledge of early, and presumably basal, members of these clades.

Latitudinal biogeography and vicariance

The deep split within Chasmosaurinae provides support for the hypothesis of latitudinal differences (but critically, not endemism) of North American Campanian dinosaur faunas, implying vicariance in the middle or (more likely) early Campanian which split chasmosaurines into a northern Chasmosaurus clade, and a southern Pentaceratops clade. Geological and biological evidence demonstrate that geographic isolation of northern and southern populations was not of continuous duration, with northern and southern biomes overlapping or mixing again by the middle Campanian.

In a series of papers, Lehman (1987, 1997, 2001; Lehman, McDowell & Connelly, 2006) proposed that in the Campanian and Maastrichtian of the North American Western Interior, dinosaur faunas were segregated into northern and southern biogeographic provinces, with the dividing line positioned roughly in central Utah. This hypothesis was criticized and partly falsified as many of the purportedly coeval northern and southern taxa were not contemporaneous and were therefore indicative of stratigraphic rather than geographic segregation (Fowler, 2006; Sullivan & Lucas, 2006; Fowler, 2017). Despite this, an expansion of Lehman’s hypothesis was proposed (Sampson et al., 2010), based partly on the description of new chasmosaurine taxa Kosmoceratops richardsoni and Utahceratops gettyi from the Kaiparowits Formation, Utah. Later (Sampson et al., 2013), previous stratigraphic criticism of the biogeographic hypothesis was rejected, suggesting that recalibrated radiometric dates (Roberts et al., 2013) showed that chasmosaurines from the Dinosaur Park Formation, Alberta and Kaiparowits Formation, Utah were indeed contemporaneous, and indicative therefore of intracontinental endemism. However, 11 out of 18 of these radiometric recalibrations of (Roberts et al., 2013) are miscalculated, some by as much as a million years (Fowler, 2017). Correctly recalibrated dates (Fowler, 2017), show the Kaiparowits taxa are stratigraphically slightly younger than the more basal chasmosaurines from Alberta, with K. richardsoni the youngest and most derived member of the Chasmosaurus lineage, and U. gettyi the oldest and most basal member of the Pentaceratops lineage. Thus the contemporaneity required for basinal-scale faunal endemism collapses.

Nevertheless, amidst this criticism, the emphasis on “lineage-thinking” in the current analysis provides evidence for a subtle form of gradational latitudinal provincialism, but not endemism. Although the Chasmosaurus and Pentaceratops lineages are not exclusive (i.e., endemic) to either north or south (a similar point is raised by both Wick & Lehman, 2013; Longrich, 2014), it is apparent that relative abundance varies latitudinally in Campanian-aged units (albeit based on a small sample size). Specimens of the Chasmosaurus clade are much more abundant in the northern United States and Canada, with the southernmost representative (Kosmoceratops richardsoni), represented by two specimens from the Kaiparowits Formation of southern Utah. Specimens of the Pentaceratops clade are more common in the southern states of New Mexico and Utah, with only one or two possible representative specimens from southern Alberta (see discussion on Chasmosaurus russelli in Supporting Information 1). This biogeographic pattern does not represent endemism as the two lineages overlap geographically during the uppermost part of the middle Campanian in Alberta and Utah. However, it is suggestive that latitudinally aligned vicariance might have been the cause of the speciation event that created the two chasmosaurine lineages. As the oldest member of the Chasmosaurus lineage occurs at ~77 Ma (see above) then vicariance must have occurred before this time. Similarly, as both lineages are seen to coexist in the uppermost part of the Dinosaur Park Formation (~76 Ma) then any physical barrier must have been passable by this time. The location of the barrier is suggested by the fact that the dividing line between northern and southern provinces appears to lie somewhere between southern Utah and northern Montana.

It has been stated (Sampson et al., 2010, 2013) that there is currently no evidence for a physical barrier separating northern and southern provinces, but this is not the case. In 1990, Lillegraven and Ostresh (not referenced by Sampson et al., 2010, 2013) produced 33 maps illustrating Late Cretaceous transgression and regression of the western shoreline of the Western Interior Seaway (WIS). The maps were at a very high stratigraphic resolution, documenting almost every ammonite zone from the middle Santonian (Clioscaphites choteauensis; 85.23 Ma; Ogg, Hinnov & Huang, 2012) through to the K-Pg boundary (66 Ma). Most importantly, the maps contrast the paleoshoreline with the modern position of the eastern Sevier thrust front of the Rocky Mountains. Although the position of the thrust front was slightly more western in the Late Cretaceous (and the mountains were not as elevated; DeCelles, 2004), it is a good approximation for the position of the upland or mountainous area which flanked the coastal plain. From these maps it can be readily observed that during the middle Santonian (85 Ma) through to the earliest part of the middle Campanian (81 Ma), the shoreline of the WIS intermittently abutted the thrust front of the incipient Rockies from central Utah to southern Alberta. For hundreds of miles the coastal plain would have been extremely narrow, in some places perhaps as little as 5–10 kilometers, providing very limited habitat. This would be similar to, for example, the modern day Zagros Mountains of Iran which are abutted by the eastern shoreline of the Persian/Arabian Gulf. This bottlenecking of the available coastal plain effectively cut off the north-south dispersal route, latitudinally bisecting the coastal plain habitat of North America into southern and northern areas separated by hundreds of miles. The latitudinal climate gradient might have exacerbated difference in local environmental conditions between northern and southern regions, although the latitudinal climate gradient was not as strong in the Late Cretaceous as it is today. Lillegraven & Ostresh (1990) show that from the early part of the middle Campanian (~80 Ma) regression of the WIS results in a broader coastal plain, and it is hypothesized here that this may no longer have presented a physiographic boundary, thereby permitting interspersal of chasmosaurine lineages, as evidenced by the presence of Pentaceratops lineage taxa in the uppermost Dinosaur Park Formation, ~76 Ma (Longrich, 2014), and later Anchiceratops in the Horseshoe Canyon Formation, ~71 Ma (Mallon et al., 2011).

The role of heterochrony in evolution of the frill and effects on phylogenetic analysis

The process of heterochrony describes changes in the rate and timing of development between stratigraphically successive populations. Most morphological trends recognized in this study are potentially controlled or affected by heterochrony, but inference of this requires knowledge of change through both ontogeny and stratigraphy. Although stratigraphic position is at least roughly known for most species in the current study, few especially young or old individuals of relatively basal chasmosaurines have been published, such that their ontogenetic change is not well understood. Nevertheless, some possible heterochronic trends can be identified or hypothesized based on the limited available material and comparison to the well documented growth series of the Late Maastrichtian derived chasmosaurine Triceratops (Horner & Goodwin, 2006, 2008; Scannella & Horner, 2010). This may have important practical implications for taxonomy and the way specimens are coded for phylogenetic analysis, but also in a broader sense may be informative about some of the unusual features of basal and derived chasmosaurines.

Development of the median embayment

The median embayment of the parietal posterior bar successively shallows and broadens from basal through derived members of the Chasmosaurus lineage, and deepens then closes in the Pentaceratops lineage. There is some evidence to suggest that similar patterns are observed ontogenetically. In “Chasmosaurus russelli”, referred adult specimen CMN 2280 has a shallow central embayment with lateral rami at an angle of 131°. The immature referred specimen, AMNH 5656, has an embayment that is less shallow (99°) and is more restricted to the central third of the posterior bar. Adult specimens of the more derived C. belli, C. sp. and Vagaceratops irvinensis have an even shallower embayment than adult “C. russelli” suggesting peramorphosis in the Chasmosaurus lineage.

Concerning basal members of the Pentaceratops lineage, there are no published juvenile specimens which preserve the median embayment, that have been recovered from the same strata as the various holotypes (and as such, could be more reliably assigned to a given taxon). Consequently, the progressive deepening of the median embayment (observed stratigraphically and phylogenetically) cannot currently be assessed for an ontogenetic component.

Development of parietal fenestrae

In Ceratopsia, the parietal fenestrae open during ontogeny by resorption of central regions of the previously solid parietal. Although this is still controversial (e.g., Farke, 2011), opening of fenestrae through ontogeny has been proposed in both basal neoceratopsians (Protoceratops; Fastovsky et al., 2011) and the highly derived Late Maastrichtian ceratopsid Triceratops (Scannella & Horner, 2010). As such, it is probable that ontogeny influences the size and shape of parietal fenestrae in both the Chasmosaurus and Pentaceratops lineages, reflected in the width of the median, posterior and lateral bars.

In adult specimens of basal chasmosaurines, the median bar of the parietal either lacks lateral flanges that invade the fenestrae, or they are only weakly developed. Flanges are more strongly developed and conspicuous in Chasmosaurinae sp. taxon C (NMMNH P-33906) and more derived chasmosaurines like Anchiceratops. It is likely that development of the flanges occurs by paedomorphosis; i.e., that flanges form as a result of the fenestrae opening less extensively during ontogeny (in more derived forms), rather than the flanges growing laterally from the median bar. It is expected therefore that juveniles of some of the more derived Pentaceratops lineage taxa (e.g., Terminocavus or taxon C) would exhibit relatively wider median bars with more developed lateral flanges, and smaller parietal fenestrae. In this respect, they might appear more similar to adults of derived chasmosaurines. This is seen in the Chasmosaurus lineage, where juvenile “C. russelli” referred specimen AMNH 5656 has very weak lateral flanges on the median bar, whereas in more mature specimens (e.g., CMN 2280) lateral flanges are absent.

The development of the broad plate-like posterior bar (in Pentaceratops lineage) and lateral bars of the parietal is similarly expected to be a result of paedomorphosis. The posterior bar of immature aff. Pentaceratops sp. SDMNH 43470 comprises a bar-like posterior portion (typical of more basal members of the Pentaceratops lineage) which has small thin flanges extending anteriorly into the parietal fenestrae. These could be interpreted as remnants of a previously more extensive plate-like part of the posterior bar that is resorbed by adulthood in more basal chasmosaurines (thereby increasing the size of the fenestrae). Hypothesized paedomorphosis in more derived members of the Pentaceratops lineage might lead to retention of this flange.

In derived chasmosaurines (e.g., “Torosaurus”, Anchiceratops, and Kosmoceratops), the lateral bars of the parietal are mediolaterally broad and completely enclose the fenestrae within the parietal. In basal chasmosaurines the lateral bars are much narrower and might not fully enclose the fenestra (such that the squamosal forms part of the lateral margin). Within the Chasmosaurus lineage, “Chasmosaurus russelli” referred adult specimen CMN 2280 is illustrated by Godfrey & Holmes (1995) as exhibiting incomplete lateral rami (i.e., the squamosal contributes to the fenestra), whereas in immature referred specimen AMNH 5656, the lateral bars are continuous, fully enclosing the fenestrae. This limited sample suggests that ontogenetic expansion of the parietal fenestrae may cause resorption of the central parts of the lateral bars, causing them to become discontinuous in adults. If so, this would be a paedomorphic trend as in specimens of the slightly more derived C. belli, the fenestra is enclosed entirely within the parietal (Godfrey & Holmes, 1995; although unenclosed fenestrae of C. sp. YPM 2016 suggest that perhaps there is no consistent pattern; Campbell et al., 2019). A similar paedomorphic trend is probably present in the Pentaceratops lineage where basal members have continuous but thin lateral bars, which are broad in Anchiceratops and more derived forms. This is only hypothetical as lateral bars are not preserved in Navajoceratops, Terminocavus, and “taxon C”.

Origin of Arrhinoceratops and the Triceratopsini

The description of intermediate morphotaxa between Pentaceratops and Anchiceratops has implications for the origin of Arrhinoceratops and the Triceratopsini (Ojoceratops + Eotriceratops + “Torosaurus” + Triceratops). In most phylogenetic analyses, Arrhinoceratops and the Triceratopsini are recovered as very closely related to Anchiceratops (e.g., Dodson, Forster & Sampson, 2004; Sampson et al., 2010; Longrich, 2014; and the current analysis). Since Anchiceratops and Arrhinoceratops were contemporaneous (co-occurring in the Horsethief and Morrin members of the Horseshoe Canyon Formation, Alberta; ~72.4–71.6 Ma; Eberth et al., 2013; Mallon et al., 2014) then the phylogenetic relationship illustrated in Fig. 9 requires that a speciation event splitting the two must have occurred prior to this time, but after the occurrence of the immediately basal Terminocavus (~74.7 Ma). However, taxa immediately basal to Anchiceratops do not resemble Arrhinoceratops, being generally characterized by a deep notch-like median embayment and large triangular epiparietals, neither of which are observed in Arrhinoceratops at any ontogenetic stage (Mallon et al., 2014). It is possible that character states shared between Arrhinoceratops and Anchiceratops (for example, small circular parietal fenestrae) may be homoplastic rather than synapomorphic, and could instead reflect shared long term trends observed across Chasmosaurinae (see above). Although this is speculative, candidates for a different origin of Arrhinoceratops and the Triceratopsini are present in the poorly known Coahuilaceratops (Loewen et al., 2010) and “Bravoceratops” (Wick & Lehman, 2013; see Supporting Information 1), from the lower Maastrichtian of Mexico and Texas, respectively. Although both taxa are known from only very scant remains, both exhibit anteriorly positioned nasal horns and retain bumps on the anterior end of the parietal relatively late in ontogeny (see Supporting Information 1): both features characteristic of Triceratopsini. Recovery of more complete specimens of Coahuilaceratops and “Bravoceratops” may be enlightening.

Regardless of their precise phylogenetic origin, the slightly embayed, cardioid shape of the frill in some specimens referred to “Torosaurus” (YPM 1831; TMM 41480-1) and Triceratops horridus (e.g., AMNH 5116) may be a remnant feature of their ancestry; a plesiomorphy or atavism exhibited by a few members of the population, which is gradually being lost. This is supported by the fact that very few specimens of Triceratops prorsus exhibit any parietal midline embayment, despite many specimens having been collected.

Conclusions

Description of the new taxa Navajoceratops sullivani and Terminocavus sealeyi, and the fragmentary Taxon C, provides critical stratigraphic and morphologic links between the Campanian Pentaceratops, and the Maastrichtian Anchiceratops, reinstating the phylogenetic hypothesis originally postulated by Lehman (1993, 1998). Combined with significant revision of other chasmosaurine taxa, this reveals a deep split of the Chasmosaurinae into Chasmosaurus and Pentaceratops clades. Morphological divergence from similar basal forms suggests the clades diverged from a common ancestor probably in the early Campanian.

Analysis of paleogeographic maps suggest that high sea level in the Santonian through to middle Campanian may have acted as an agent of vicariance, separating an ancestral chasmosaurine population into northern and southern subpopulations which over time led to divergence and speciation. This lends support to recent hypotheses of latitudinally arrayed differences in terrestrial faunal composition (e.g., Lehman, 1987, 1997, 2001), but stops short of supporting basinal-level endemism in the middle to late Campanian (e.g., Sampson et al., 2010).

Description of the new material places San Juan Basin chasmosaurines as among the best documented of their clade, second only to Triceratops in number of specimens and quality of accompanying data.

Although this work presents significant revision of many chasmosaurine taxa, much reanalysis and redescription remains. Inclusion of more recently described taxa and separation of problematic taxa and specimens (see Supporting Information 1) will be attempted in forthcoming manuscripts based on Fry (2015) and Fowler & Freedman Fowler (2017).

Supplemental Information

Supplemental Information 1 Review of specimens and taxa relevant to main analysis, taxonomy, morphology, stratigraphy, ontogeny.

Additional materials and methods including geological setting, and review of fossil materials, including discussion of specimen morphology.

Click here for additional data file.

Supplemental Information 2 Phylogenetic analysis : character descriptions, revisions, and new characters.

A detailed review of morphological characters used in the phylogenetic analysis. Various revisions are offered to the base character matrix of Mallon et al. (2014), itself based on Sampson et al. (2010). Seven characters are revised, and four new characters are added.

Click here for additional data file.

Supplemental Information 3 Hypothesis of chasmosaurine relationships from Lehman (1998).

Lehman (1998) proposed that Campanian chasmosaurines belonged to two lineages evolving divergently from initially similar morphology. A Chasmosaurus lineage (left, C. russelli 1-3, C. belli 4-7) was characterized by progressive shallowing of the parietal median embayment, development of epiparietals into an elongate ridge at locus ep1, and lateral migration of loci ep2 and 3. In contrast, an Agujaceratops (8) - Pentaceratops (9, 10) - Anchiceratops (11, 12) lineage (right) was characterized by a deepening median embayment which caused rotation of epiparietals at locus ep1 to form the butterfly-wing orientation characteristic of Anchiceratops. Lehman’s hypothesis was consistent with the stratigraphic distribution of the depicted specimens, and was further supported by the subsequent discovery of new taxa. Facsimile of Lehman (1998).

Click here for additional data file.

Supplemental Information 4 Alternative hypotheses of chasmosaurine phylogeny.

Two recent phylogenetic analyses contrast the dual-lineage hypothesis of Lehman (1998). A, Sampson et al. (2010) recover a phylogeny where Pentaceratops is unrelated to Anchiceratops; where Vagaceratops is unrelated to Chasmosaurus; and where instead Kosmoceratops and Vagaceratops form an outgroup to Anchiceratops and all other more derived chasmosaurines. B, Mallon et al. (2014; drawing on the same phylogenetic matrix as Sampson et al., 2010) published a phylogeny where the Lower Maastrichtian taxa Anchiceratops and Arrhinoceratops occur in a basal polytomy, and some of the stratigraphically oldest taxa form the most derived clade (Middle to Upper Campanian C. belli + C. russelli). These new analyses require significant ghost lineages be present for most clades, for which there is currently no fossil evidence. A, B adapted from Sampson et al. (2010) and Mallon et al. (2014) respectively.

Click here for additional data file.

Supplemental Information 5 Historical stratigraphic terminology of the Fruitland and Kirtland Formations.

The Fruitland and Kirtland Formations of the San Juan Basin have undergone many changes regarding terminology, and (more importantly) the definitions of member or formational contacts. Here I illustrate all the revisions in chronological order, ending with the current terminology and definitions used in this study. Adapted from Bauer (1916); Baltz, Ash & Anderson (1966); Fassett & Hinds (1971); Hunt & Lucas (1992, 2003); Sullivan, Lucas & Braman (2005) and Lucas et al. (2006). Section thickness in meters (m).

Click here for additional data file.

Supplemental Information 6 Navajoceratops sullivani holotype SMP VP-1500 parietal exposed before collection and preparation.

(A) An area of rugose concreted bone occurred at locus ep1 and extended around the anterior border of the median embayment (em), but was mostly unrecoverable during preparation. This is interpreted here representing ep1. (B) The median bar (mb) of SMP VP-1500 was slightly displaced laterally and ventrally. ep1, epiparietal 1. f, parietal fenestra. L-lr / R-lr, Left / Right lateral rami of the posterior bar. Hammer for scale = 28 cm.

Click here for additional data file.

Supplemental Information 7 Navajoceratops sullivani holotype SMP VP-1500 isolated frill epiossification.

Isolated frill epiossification found with parietal. Possibly an epiparietal, although squamosal fragments were found weathered out on the surface. Scalebar = 5 cm.

Click here for additional data file.

Supplemental Information 8 Navajoceratops sullivani holotype SMP VP-1500 squamosal fragments.

Two of the largest fragments of the squamosal found weathered on the surface next to the parietal. A (dorsal), B (ventral; inferred orientations), squamosal fragment exhibiting two fused episquamosals (es). C, D, elongate fragment, possibly from the posterior end of the squamosal where it narrows. Both fragments exhibit characteristic ceratopsian vascular surface texture. Scalebar = 10 cm.

Click here for additional data file.

Supplemental Information 9 Navajoceratops sullivani holotype SMP VP-1500 jugal.

Possible ventralmost end of a jugal with fused epijugal (ej), in lateral and medial views (inferred). Alternatively, this may be an episquamosal fused to a small part of the squamosal. Scalebar = 5 cm.

Click here for additional data file.

Supplemental Information 10 Terminocavus sealeyi holotype NMMNH P-27468 right squamosal.

Ventral (left) and dorsal (right) views. sb, squamosal bar. es, episquamosal. Scalebar = 10 cm.

Click here for additional data file.

Supplemental Information 11 Terminocavus sealeyi holotype NMMNH P-27468 left jugal.

Ventral two thirds of left jugal with fused epijugal (ej) and partial quadratojugal in posterior (A), left lateral (B) and Anterior (C) views. A size comparison of NMMNH P-27468 (D) is compared with Utahceratops gettyi referred specimen UMNH VP-12198 (E). Scalebars = 10 cm.

Click here for additional data file.

Supplemental Information 12 Angle formed by the lateral rami of the parietal posterior bar in Pentaceratops and close relatives.

In most chasmosaurines the lateral rami of the parietal posterior bar meet medially at an angle, forming an embayment. Specimens are shown here in stratigraphic order where possible. (A) is probably the stratigraphically oldest specimen illustrated (see supp. info. text). The stratigraphically separated taxa Utahceratops (B), Pentaceratops (C, D, F), Navajoceratops (G), to Terminocavus (H) form a morphologic spectrum, recording overall decrease in the angle of the lateral rami of the posterior bar, deepening and narrowing the median embayment. Agujaceratops specimen UTEP P.37.7.065 SDMNH 43470 (E) is of uncertain stratigraphic position, but may be roughly equivalent to the Hunter Wash Member of the Kirtland Formation, New Mexico, from which Navajoceratops (G) and Terminocavus (H) were collected. Specimens not shown to scale (see main text figure for relative sizes).

Click here for additional data file.

Supplemental Information 13 Angle formed by the lateral rami of the parietal posterior bar in Chasmosaurus and close relatives.

In most chasmosaurines the lateral rami of the parietal posterior bar meet medially at an angle, forming an embayment. Taxa are illustrated in stratigraphic order where possible. The stratigraphically separated taxa "Chasmosaurus russelli" (B-D), C. belli (E-J), Vagaceratops irvinensis (K), and Kosmoceratops richardsoni (H) form a morphologic spectrum, recording overall increase in the angle of the lateral rami of the posterior bar, shallowing the median embayment. Specimens not shown to scale.

Click here for additional data file.

Supplemental Information 14 Selected measurements of Navajoceratops sullivani holotype SMP VP-1500.

Parietal shown in dorsal view. Measurements in cm (1.d.p).

Click here for additional data file.

Supplemental Information 15 Selected measurements of Terminocavus sealeyi holotype NMMNH P-27468.

Parietal shown in dorsal view. Measurements in cm (1.d.p).

Click here for additional data file.

Supplemental Information 16 Selected measurements of Terminocavus sealeyi holotype NMMNH P-27468.

Jugal - epijugal shown in posterior (left) and left lateral (right) views. Measurements in cm (1.d.p).

Click here for additional data file.

Supplemental Information 17 Previously undescribed specimen, aff. Pentaceratops n.sp., NMMNH P-37880.

NMMNH P-37880, a partial right lateral ramus of parietal posterior bar in posterior, dorsal, medial, and ventral views. Although an isolated skull fragment, the posterior bar of the parietal is the most diagnostic element in Campanian chasmosaurines. Specimen recovered from the Fossil Forest Member, Fruitland Formation (San Juan Basin, New Mexico) and is morphologically most similar to other specimens referred to aff P. n. sp. Abbreviations: em, median embayment of the posterior bar; ep, epiparietal loci numbered by hypothesized position (no epiossifications are fused to this specimen). f, parietal fenestra. Scalebar equals 10 cm. Reconstruction line drawing based on c.f. P. sternbergii specimen UKVP 16100.

Click here for additional data file.

Supplemental Information 18 Reidentification of parietal median bar of Bravoceratops polyphemus (Wick & Lehman, 2013).

The parietal fragment of TMM 46015-1 (Bravoceratops polyphemus, A) is reidentified here as representing the anterior half of the parietal median bar as it compares favorably with the anterior median bars of c.f. Pentaceratops (USNM 8604, B; PMU 24924, C) and Chasmosaurus belli (CMN 491; D). The specimen was previously identified as the posterior half of the parietal median bar by Wick & Lehman (2013, E), and interpreted as such bore most of the morphological features which distinguished this new taxon. Scalebars equal 10 cm. No scale available for PMU 24924 (C). (A, E) adapted from Wick & Lehman (2013); (B) adapted from Gilmore (1919); (C) adapted from Wiman (1930); (D) adapted from Hatcher, Marsh & Lull (1907).

Click here for additional data file.

Supplemental Information 19 "Pentaceratops aquilonius" (CMN 9813) shown to scale with c.f. P. sternbergii (AMNH 1625).

Longrich (2014) proposed the new taxon Pentaceratops aquilonius (holotype CMN 9813, A) based in part on the suggestion that compared to c.f. Pentaceratops sternbergii (AMNH 1625, B), the parietal posterior bar is anteroposteriorly broad and only weakly embayed. This is based on inaccurate reconstruction of CMN 9813, mainly because of inappropriate scaling. This figure shows CMN 9813 is comparable to AMNH 1625 in the anteroposterior thickness of the posterior bar, and that CMN 9813 is too incomplete to infer size or shape of the median embayment. (A, B) Adapted from Longrich (2014). Scalebar equals 10 cm.

Click here for additional data file.

Supplemental Information 20 Proposed reconfiguration of epiparietal numbering system in Chasmosaurus and related chasmosaurines.

The original epiparietal numbering systems of Holmes et al. (2001; A, B) and Sampson et al. (2010, C) are reconfigured (G-I) based on comparison to stratigraphically preceding chasmosaurines c.f. Chasmosaurus russelli (CMN 2280, D), and C. belli (ROM 843, E; YPM 2016, F). Most notably, locus ep1 develops from an spindle shaped epiparietal in c.f. C. russelli (D) to an elongate and anteriorly curving ridge in C. belli (E, F). In YPM 2016 the ep1 ridge bears 3-4 anteriorly projecting processes which are here interpreted to be homologous with the anteriorly projecting processes also in this position in the stratigraphically succeeding taxa Vagaceratops (Chasmosaurus) irvinensis (G, H) and Kosmoceratops richardsoni (I). Specimens not shown to scale.

Click here for additional data file.

Supplemental Information 21 Angle of postorbital horn relative to maxillary tooth row in Chasmosaurinae sp. "Agujaceratops mariscalensis" TMM 43098-1 (A) and Pentaceratops sternbergii AMNH 6325 (B).

In their rediagnosis of Agujaceratops mariscalensis, Forster et al. (1993) suggest "erect supraorbital horncores that attain an angle of 85° to the maxillary tooth row in adults" as a new autapomorphy, based on TMM 43098-1 (A). This figure shows that the postorbital horns of TMM 43098-1 are no more erect than those of Pentaceratops sternbergii holotype AMNH 6325 (B). The posterior curvature of the postorbital horns in TMM 43098-1 create the illusion of more erect horns. (A) adapted from Forster et al. (1993); (B) adapted from Osborn (1923).

Click here for additional data file.

Supplemental Information 22 Phylogenetic analysis.

Character matrix for phylogenetic analysis. Results of analysis given in main manuscript. Character discussion and other information provided in Supporting Information 2.

Click here for additional data file.

Special thanks to John R. Horner and Robert M. Sullivan for immeasurable support and opportunity. Thanks for specimen access or assistance, discussion, or general help to Arjan Boere, David Eberth, David Evans, Joshua Fry, David Gillette, Andrew Heckert, Steven Jasinski, Benjamin Kear, Matt Lavin, Spencer Lucas, John Scannella, Paul Sealey, Justin Spielmann, Steven Wick, Tim Williams, John Wilson, and Oliver Wings. Thanks to the Bureau of Land Management for collection permits issued to Robert M. Sullivan and Thomas Williamson. Helpful reviews kindly provided by James Campbell, Mark Loewen, and Jordan Mallon.

Institutional Abbreviations

AMNH American Museum of Natural History, New York City, New York, USA

CMN (formerly NMC) Canadian Museum of Nature, Ottawa, Ontario, Canada

MNA Museum of Northern Arizona, Flagstaff, Arizona, USA

NMMNH New Mexico Museum of Natural History and Science, Albuquerque, New Mexico, USA

OMNH Oklahoma Museum of Natural History, Norman, Oklahoma, USA

PMU Paleontologiska Museet, Uppsala University, Uppsala, Sweden

SDNHM San Diego Natural History Museum, San Diego, California, USA

SMP State Museum of Pennsylvania, Harrisburg, Pennsylvania, USA

KUVP University of Kansas, Lawrence, Kansas, USA

UMNH Utah Museum of Natural History, Salt Lake City, Utah

UNM University of New Mexico, Albuquerque, New Mexico, USA

USNM United States National Museum, Smithsonian Institution, Washington D.C., USA

UTEP University of Texas at El Paso, El Paso, Texas, USA

Additional Information and Declarations

Competing Interests

Author Contributions

Data Availability

New Species Registration

The authors declare that they have no competing interests.

Denver W. Fowler conceived and designed the experiments, performed the experiments, analyzed the data, prepared figures and/or tables, authored or reviewed drafts of the paper, and approved the final draft.

Elizabeth A. Freedman Fowler performed the experiments, analyzed the data, prepared figures and/or tables, and approved the final draft.

The following information was supplied regarding data availability:

Phylogenetic analysis data:

Description of new phylogenetic characters and coding of specimens is available in File S1. The character matrix for phylogenetic analysis is available in Data S1.

Newly described specimen locations:

SMP VP-1500 is stored in the federal repository of the State Museum of Pennsylvania, Harrisburg, PA.

NMMNH P-27488; NMMNH P-33906; and NMMNH P-37880 are stored in the federal repository of the New Mexico Museum of Natural History and Science, Albuquerque, NM.

The following information was supplied regarding the registration of a newly described species:

Publication LSID:

urn:lsid:zoobank.org:pub:E2ECA33C-63A8-4EFF-9EB4-BCF7ED28C63E.

Navajoceratops sullivani LSID:

urn:lsid:zoobank.org:act:038D3DF1-DB41-48AF-9791-14C846971133.

Terminocavus sealeyi LSID:

urn:lsid:zoobank.org:act:6E5A8D79-1F2C-484F-BED7-7C556C5C062A.

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
