# Peer review of "Transitional evolutionary forms in chasmosaurine ceratopsid dinosaurs: evidence from the Campanian of New Mexico"

_PeerJ, doi:10.7717/peerj.9251_

## Round 0.1 · original submission · Major Revisions

Dear Denver,

I have now received three reviews of your manuscript. On this basis and the fact that I personally do not find your hypothesis of anagenesis well supported, I recommend major revisions.

Please, together with your unmarked revised manuscript, provide a marked-up copy as well as a document explaining how you have addressed each of the points raised by the reviewers. I will then send the new version out again for review to Reviewers 1 and 2.

Best regards,
Fabien Knoll

·

Basic reporting

No comment.

Experimental design

No comment.

Validity of the findings

No comment.

Additional comments

I very much enjoyed reading the authors’ paper describing new chasmosaurine taxa from the Campanian of New Mexico, and stratigraphic trends in chasmosaurine clades over time, and applaud the authors' hard work and efforts. The new taxa are stratigraphically and morphologically intermediate to Pentaceratops and Anchiceratops, and the authors make a compelling argument that this represents morphological evolution over time. This paper contributes greatly to our understanding of chasmosaurines in Laramidia and provides several new avenues of research. The paper is well-written, organized, and illustrated, and it was an absolute pleasure of mine to review it.

I do have some minor critiques, which should be addressed before acceptance. I have included an annotated version of the manuscript, showing my comments in yellow sticky-notes. I also have some additional comments on the supplementary material (see below).

My general critique is that the stratigraphic trends outlined by the authors for chasmosaurines in the Dinosaur Park Formation (Chasmosaurus belli, C. russelli, and Vagaceratops irvinensis) are too simplified, and do not reflect the stratigraphic positions of specimens as given in Campbell et al. (2016, 2019). In Campbell et al. (2019), I assign two specimens (YPM 2016 and TMP 2009.034.0009) as basal members of the Vagaceratops lineage, and these specimens lie stratigraphically below some or all C. belli specimens (ROM 843 and CMN 2245), indicating that these two taxa arose via cladogenesis, not anagenesis as the authors propose. Also, the gradual lateral expansion of the epiparietal 1 locus moving from C. russelli to C. belli, as proposed by the authors, is not supported by the stratigraphy of specimens, as CMN 2245 (C. belli) has a modest-sized, unadorned epiparietal 1 locus, and occurs stratigraphically above ROM 843 (C. belli), which has a laterally-expanded epiparietal 1 locus. I agree with the authors that C. russelli, C. belli, and V. irvinensis are all closely-related , but the stratigraphic data suggests a cladogenetic (not anagenetic) origin for these taxa.


Below are my comments for the supplemental material.

Figs. S10, S13, S14:
“sealyi” should read “sealeyi”

Fig. S11:
“BMNH 4948” should read “NHMUK R4948”

Fig. S16:
“NMC 491” should read “CMN 491”

Fig. S18:
I agree with the authors’ interpretation (Supp. Info. 1 file) that the epiparietal on the posterolateral corner of the parietal in specimens that have more than 3 epiparietals (e.g., epiparietal 5 in Vagaceratops, YPM 2016, and Kosmoceratops) is most likely evolutionarily derived from the epiparietal 3 seen in more basal taxa that only have 3 epiparietals; and that the only reason why the posterolateral corner epiparietal in Vagaceratops and YPM 2016 is referred to as “epiparietal 5” and not “epiparietal 3” is simply because additional epiparietals have subsequently appeared between it and the midline of the parietal. In light of this, I understand the authors’ point-of-view in wanting to designate “epiparietal 3” as the epiparietal that occupies the posterolateral corner of the frill, regardless of how many epiparietals are visible on a given specimen.
I can also understand the authors’ point-of-view that these two additional epiparietals may have emerged via the elongation of the epiparietal 1 locus, as some C. belli specimens (e.g., CMN 491, ROM 843) have what appears to be a laterally extensive epiparietal 1 locus. However, looking at the stratigraphy of only C. belli specimens (see Campbell et al., 2016), the epiparietal 1 locus does not increase laterally over time, as ROM 843 occurs below CMN 2245, the latter of which has only a modest-sized, unadorned epiparietal 1 locus (see Fig. 16 of Campbell et al., 2016). Also, the stratigraphic trend proposed by the authors, wherein C. belli is replaced stratigraphically with V. irvinensis, is not supported by the stratigraphic data presented in Campbell et al. (2016, 2019). YPM 2016 and TMP 2009.034.0049 – specimens identified by Campbell et al. (2019) as being basal members of the Vagaceratops lineage – occur below either all C. belli specimens (in the case of YPM 2016) or some C. belli specimens (in the case of TMP 2009.034.0009). As the authors point out in Supp. Info 1 file, there is some uncertainty over the stratigraphic position of YPM 2016. This uncertainty is addressed by Campbell et al. (2019), who say that the Bearpaw Formation (and Lethbridge Coal Zone) are not exposed anywhere near the site that yielded YPM 2016, so the specimen’s purportedly close stratigraphic position below those horizons (as written in the original field notes) appear somewhat dubious; instead, Campbell et al. (2019) consider the quarry identified in the 1930’s (which is located in a geographic location consistent with the original field notes) as being the YPM 2016 site, which occurs much lower in section. But, apart from YPM 2016, the stratigraphic overlap of C. belli and the Vagaceratops lineage is further supported by the relatively low stratigraphic position of TMP 2009.034.0009 relative to the C. belli specimen CMN 2245.
I confess that I haven’t seen Kosmoceratops skulls in-person, so I can’t comment on the location of the parietal-squamosal suture. However, I think that the authors’ placement of this suture (i.e., such that each side of the parietal has 5 epiparietals and not 3, as originally proposed by Sampson et al., 2010) is much more convincing than that described by Sampson et al. (2010), who concluded that the squamosal wrapped around the back of the frill, which I’ve always had trouble believing.
As for the origin of the two additional epiparietals in Vagaceratops, YPM 2016, and Kosmoceratops, the loci underlying the epiparietals in these specimens are admittedly obscured, making it difficult to say whether the medialmost 3 epiparietals rest on separate loci, or one continuous, elongated epiparietal 1 locus. I believe that AMNH 5402 can shed some light on this. In this specimen, three epiparietals are present on each side, occupying homologous positions as epiparietals 1, 4 and 5 in Vagaceratops (Campbell et al., 2019). Epiparietal 1 in this specimen sits on its own, low-relief, comparatively-sized locus, and between epiparietals 1 and 4 lie two additional, low-relief loci, occupying homologous positions as epiparietals 2 and 3 in Vagaceratops. So, if AMNH 5402 represents a basal member of the Vagaceratops lineage (as I interpret in Campbell et al., 2019), this would indicate that the 3 medialmost epiparietals in Vagaceratops are attached to their own, separate loci.

·

Basic reporting

This paper is generally well-written, using clear language throughout, although I have numerous grammatical suggestions (and other considerations) that are given in the attached ms markup. Covering as much ground as it does (naming of two new species, chasmosaurine taxonomy and phylogeny, heterochrony, and biogeography), the paper is really quite long, and although I typically appreciate more detail than less, I found it a difficult read because I was not always able to follow all of the different threads of discussion through the manuscript. Since the two newly erected species have bearing on the authors' proposed anagenetic lineage from Pentaceratops to Anchiceratops, my strongest suggestion might be that the authors focus on this topic, rather than trying to address every detail of chasmosaurine phylogeny and biogeography, which are better saved for a separate paper.
The manuscript also suffers from not being current with the literature (likely a result of fact that this paper stems from a dissertation written some years ago). A few omissions stand out: (1) James Campbell's recent work on the Chasmosaurus belli-irvinensis transition, which contradicts the hypothesis of anagenesis between these species (Campbell et al. 2019. Vertebrate Anatomy Morphology Palaeontology 7:83-100); (2) my own work on the status of Pentaceratops "aquilonius" (Mallon et al. 2016. PLOS ONE 11:e0154218); (3) a couple of recent papers on ontogeny in non-Triceratops chasmosaurines (Mallon et al. 2015. Zoological Journal of the Linnean Society 175: 910-929; Currie et al. Journal of Vertebrate Paleontology 36:e1048348). I would also strongly discourage Fowler and Freedman-Fowler from citing the Longrich F1000 manuscript. This is an online pre-print that was never formally published because it did not pass peer review. As such, the taxa named therein (and repeated here; e.g., Chasmosaurus "priscus") are invalid.

Experimental design

I think the most serious issues with this manuscript are with the experimental design.
First, I am uncertain about the application of geometric morphometrics here. The authors have decided a priori that the most interesting variation is in the parietal embayment and epiparietal orientation (something I don't necessarily disagree with). Therefore, they design their analysis to capture strictly these sources of variation and find that... their specimens differ in those very features they chose to look at. This approach seems almost tautological to me, unless I am missing something, in which case, I hope the authors will clarify. I think simply illustrating the frills in stratigraphic succession is enough to make the authors' point re: the linked changes in frill morphology. The abstraction of (half) frill shape through landmarks unnecessarily confuses the matter.
A still bigger issue is with the phylogenetic analysis. It's really obvious from the outset that Fowler and Freedman-Fowler have decided a priori on their phylogeny of choice, which is one divided into clades leading to Kosmoceratops, Anchiceratops, and Triceratops. And yet, when their phylogenetic analysis rejects their preferred topology (contrary to what they state in lines 877-879, but consistent with what they state in lines 1115-1116), the authors effectively ignore the results and blame the working matrix for being inadequate. To be sure, I agree that there are issues with the current working matrix, but if the authors are going to basically ignore the outcome of their analysis because it does not support their pet hypothesis, then I don't see the point of doing a cladistic analysis at all.
A couple of other issues with the cladistic analysis: If the authors are going to go there, I would be curious to know the CI and RI statistics and how these compare with published analyses. It is also disappointing to see that some taxa (e.g., Regaliceratops, Spiclypeus) were not coded into the cladistic analysis. These are both based on relatively complete skulls, and were published a few years ago now. I would love to know how they fit into the authors' preferred phylogenetic scheme, since I'm not sure they fit comfortably into any of their three envisioned clades.
Lastly, Fowler and Freedman-Fowler make the case that many of the medialmost epiparietals in Vagaceratops and Kosmoceratops are outgrowths of the ep1 locus, as evidenced by Chasmosaurus belli YPM 2016. However, this specimen was recently described by Campbell et al. (cited above), and they identify discrete epiossifications along the parietal, rather than simply "an elongate ridge" in the ep1 position. The authors need to address this disagreement because, as they note, it affects the coding of 17 characters in their cladistic analysis, which throws the whole exercise into question.

Validity of the findings

Strictly speaking, the conclusions do not follow from the results because the authors' preferred set of relationships among chasmosaurines does not follow from their own cladistic analysis. However, I'm generally on board with what the authors are trying to do, here. I accept that they have found new species. I applaud their otherwise taxonomically conservative approach. I find the hypothesis that there exists a lineage from Pentaceratops to Anchiceratops, where the parietal embayment gradually closes, intriguing. I just don't think their methodological approach is justified.
It's hard to know where to go from here. If Fowler and Freedman-Fowler think the anagenetic relationships they propose are better than the topology(-ies) returned by recent cladistic analyses, perhaps they could use cladistic modeling to determine how many more steps it would take to recover their preferred topology, and to identify which problematic characters (I'm sure there are many) are leading chasmosaurine researchers astray. Perhaps we should be weighting frill characters over, say, horncore characters? I think these approaches would be much more informative than simply running an updated working matrix and then ignoring the results.

Additional comments

I anticipate that implementing even some of my comments will require a lot of re-writing, but I think the gist of what Fowler and Freedman-Fowler are trying to do is warranted, and will help invigorate systematic investigations of Chasmosaurinae again. For this reason, I recommend publication pending major revisions.

·

Basic reporting

Flawless.

Experimental design

Perfectly acceptable.

Validity of the findings

No comment

Additional comments

I recommend acceptance of this manuscript for publication in PeerJ as is. This paper adds greatly to the discussion of chasmosaurine radiation and morphological evolution and will be welcomed by the ceratopsid working community, the paleontology community in general and the general public. I do not agree with everything the authors propose, but this paper will further discussions regarding taxonomic validity and provincialism. As far as grammar, anatomical nomenclature, figures and captions, I have never reviewed a paper that is this flawless in its first iteration. The authors are to be commended. In summary, I think this is a great paper that will generate a lot of interest in the ceratopsian worker community.

---

## Round 0.2 · Minor Revisions

Dear Denver,
Your revised manuscript has been re-evaluated by two of the original referees. Areas for improvement have been identified, which require attention.

Please, together with your unmarked revised manuscript, provide a marked-up copy as well as a document explaining how you have addressed each of the points raised by the referees in this second round of reviews.

Thank you for your attention.

Stay safe,

Fabien

·

Basic reporting

Clear and unambiguous, professional English used throughout. Literature references, sufficient field background/context provided. Professional article structure, figures, tables. Raw data shared. Self-contained with relevant results to hypotheses.

Experimental design

Original primary research within Aims and Scope of the journal. Research question well defined, relevant and meaningful. It is stated how research fills an identified knowledge gap. Rigorous investigation performed to a high technical and ethical standard. Methods described with sufficient detail and information to replicate.

Validity of the findings

All underlying data have been provided; they are robust, statistically sound, and controlled. Conclusions are well stated, linked to original research question and limited to supporting results.

Additional comments

I appreciate the changes the authors made to acknowledge differing interpretations of anatomy and stratigraphic positions of specimens, including those presented in Campbell et al. (2016, 2019). Not to belabour the opinion presented in Campbell et al. (2019), but the quarry which they believed to have yielded YPM 2016 doesn’t directly contradict the original field notes for this specimen, as the geographic location of the supposed quarry is consistent with the geographic location given in the field notes, and in that broad area of badlands the Bearpaw Formation is not exposed at all, making it most likely that YPM 2016 wasn’t collected near the base of that unit; anyways though, I’m happy to agree to disagree about the stratigraphic position of this specimen.

I also appreciate that the authors don’t rule out the possibility of cladogenesis amongst chasmosaurines from the Dinosaur Park Formation, due to possible stratigraphic overlap between different taxa. Like the authors, I believe it’s possible that a population of “C. russelli” (i.e., excluding C. russelli holotype) could give rise to C. belli, and C. belli to C. irvinensis, and possibly even C. irvinensis to Kosmoceratops richardsoni (given the new squamosal/parietal interpretation given by the authors for the latter), but given that there appears to be some stratigraphic overlap between these taxa, cladogenesis can’t be ruled out for these taxa. As the authors say, regardless of whether these taxa arose via cladogenesis or anagenesis, this doesn’t change the overall morphological changes that occur over geological time. As the authors say, more specimens are needed to better understand the stratigraphic and evolutionary relationships of these taxa.

In my first review of this paper, my understanding was that the authors suggest that the parietal locus underlying epiparietal 1 (in Chasmosaurus specimens with three epiparietals on each side of the frill) expands laterally over evolutionary time, and that two additional epiparietals appear on this ridge, for a total of five epiparietals on each side of the frill (as in Vagaceratops, YPM 2016, and AMNH 5402, and likely Kosmoceratops too as the authors suggest). In my first review, I sided with the authors’ interpretation that the large epiparietal on the posterolateral corner of taxa/specimens with five epiparietals (i.e., “epiparietal 5”) is most likely derived from the large epiparietal on the posterolateral corner of Chasmosaurus specimens with three epiparietals (i.e., “epiparietal 3”); for these reasons, I understand the authors’ interpretation of these corner epiparietals as being evolutionary homologous elements. I think this is a reasonable explanation. As the authors say, demonstrating the homology of particular epiparietals between basal chasmosaurines (e.g., Chasmosaurus) and triceratopsins is more difficult, given that the epiparietals in the latter (e.g., Triceratops) are fairly uniform in size and shape, that there is no distinct posterolateral corner of the frill (e.g., Triceratops), and there’s a median epiparietal (as in Triceratops and Regaliceratops) – perhaps this epiparietal arose from the conjoining of the medialmost pair of epiparietals?

In the original draft of the authors’ paper, my understanding was that the authors were proposing that the epiparietal 1 locus expands laterally over evolutionary time, and that additional epiparietals (i.e., “epiparietals 2 and 3” in taxa/specimens with five epiparietals) subsequently develop on top of it. In my first review, I presented a counter-argument that there appear to be separate loci for “epiparietals 2 and 3” in AMNH 5402 (Chasmosaurus sp., a specimen which Campbell et al. 2019 interpret as having loci for five epiparietals). I made this counter-argument to show that the three medialmost epiparietals (i.e., “epiparietals 1, 2, and 3”) in taxa/specimens with five epiparietals may not necessarily be borne on a single, elongate locus – which the authors were proposing. In the authors’ rebuttal letter though, the authors’ say that it doesn’t matter whether the three medialmost epiparietals are borne on a single elongate locus or on separate loci, and that the take-home part of their hypothesis is that the two additional epiparietals seen in Vagaceratops, YPM 2016, AMNH 5402, and Kosmoceratops likely developed along the part of the parietal between the midline and “epiparietal 2” of Chasmosaurus. I understand now that the authors aren’t wedded to the idea that the three medialmost epiparietals are necessarily borne on a single elongated locus, but they weren’t explicit about this in the original draft of their paper, so their “frustration” with my (and Reviewer 2’s) misunderstanding of this is a bit unfair. Anyways, I thank the authors for clarifying their stance on this.

I’m happy with the changes made by the authors, and recommend the paper for acceptance, after the following changes are made. These are all very small fixes that will improve the consistency and clarity of the paper. Points 11 through 17 are points that I made in my first review which weren’t implemented but should be.

1) “and” needs to be added to the list of authors for the following references in the references section: Currie et al. (2016), Holmes et al. (2020), Mallon et al. (2016).

2) Line 172: change “indet..” to “indet.”. Also change “sp..” to “sp.” in lines 994 and 1073.

3) Line 251: space needed within “C.russelli”.

4) Line 624: change “Figures” to “Figs.”, since in brackets. Same thing on Line 943. Also, change “Figure” to “Fig.” in lines 998, 1011, 1076, and 1132.

5) Line 954: “Anchiceratops” should be italicized.

6) Line 1242: add comma after “However”.

7) Line 1311: add comma after “Consequently”.

8) Line 1327: change “ie.” to “i.e.”

9) Line 1356: change “fenstrae” to “fenestrae”.

10) In the paragraph before the Discussion section, “Campbell (2015)” should be added to the list of references of newly described chasmosaurine taxa, since Judiceratops is rediagnosed in that paper.

11) Descriptors like “upper” “late”, etc. preceding stages (e.g., Campanian) should be lower case. Some were fixed, but not all of them.

12) Under “Institutional Abbreviations”, states/provinces are given in some cases but not in these – need to be consistent.

13) “Campbell et al. (2016)” is cited in text but needs to be added to the references section. Conversely, “Campbell et al. (2013)” appears in the references section but isn’t cited in text (please remove from references section). Also, “Campbell (2015)” needs to be added to the references section.

14) In the “Locality and Stratigraphy” section of “Navahoceratops sullivani”, still need to say that the “uppermost local coal” is the uppermost coal bed of the Nehnahnezad Member. Also, in “Diagnosis”, still need to specify which specimens you’re referring to that have parietal posterior bar angles of 87 or 88.

15) Some changes still need to be made to Figure 1. The river names are difficult to read – the font should be changed from blue to black. The rivers are also hard to see – a thicker, darker blue line would be easier to see. Also, a white square should be placed underneath the state map in the bottom left corner so that it’s distinct from the rest of the map; the state map is particularly confusing where it is since a river flows through it. Also, “Denazin” appears in the caption and the legend, yet this member is spelled “De-na-zin” elsewhere in the paper.

16) A lithology legend still needs to be added to Figure 2. Abbreviations for members also need to be included in the caption; alternatively, to save space, maybe these member names can be added to the figure itself by connecting an arrow between the stratigraphic interval and the full name beside it?

17) In Figure 10, “Bearpaw Shale” still needs to be changed to Bearpaw Fm”.

I congratulate the authors on all of their hard work and I look forward to seeing the final published version. I think this paper will generate good discussion amongst other researchers in the field, and will advance our understanding of the morphological evolution of chasmosaurines. I also thank the authors for the opportunity for me to review it.

Sincerely,

James Campbell

·

Basic reporting

See General Comments.

Experimental design

See General Comments.

Validity of the findings

See General Comments.

Additional comments

I'm generally satisfied with the changes that the authors made, although these pertain mainly to my minor concerns. I still have to respectfully (but strongly) disagree with the authors about not re-focusing their manuscript on that lineage containing the two new taxa they are describing (Pentaceratops --> Anchiceratops). The authors argue that reviewing chasmosaurine relationships more broadly provides better context, but I really think it does so at the expense of comprehension (causing the reader to miss the forest for the trees). I think that, where specific character coding issues have broader implications for chasmosaurine cladistics, those might be pointed out in the Discussion, but the reevaluation of the Chasmosaurus lineage is best explored in another focused paper (like the planned paper on the Oldman Fm Chasmosaurus). I suspect this is an issue on which we'll have to agree to disagree, and the editor and authors can do with my suggestion what they wish.

Re: the landmark analysis (which I still think isn't very helpful, but will concede the point), I did not mean to imply that the authors re-do the analysis using entire (rather than half) frills. Rather, I think it would be helpful if, instead of illustrating half frills in their Fig. 8 warp grids, the entire frill could be shown. Is this possible? Maybe using outline drawings? The grids themselves are very difficult to interpret, and could be relegated to the Appendix.

Drs. Fowler and Fowler-Freedman disagree with me about the support that their cladistic analyses offer their hypothesized relationships (Penta --> Navajo --> Termino --> Anchi), stating:
"The only part of the phylogenetic analysis that doesn't fit the lineage hypothesis is that Coahuila, Penta & Utah form their own clade, rather than forming stem taxa leading up to Navajo, Termino, and Anchi + Triceratopsini. So it's not perfect, but then the results of phylogenetic analysis are phylogenetic hypotheses:  A cladogram is not data, and if it does not match specimen data (stratigraphy, ontogeny or morphology) then it can be treated as suspect."
I agree with them that a cladogram is not data. But a cladogram is based on data (in this case, morphological data), and these data most parsimoniously unite Pentaceratops with Coahuilaceratops, Bravoceratops, and Utahceratops; not on the branch leading to Anchiceratops and beyond. The common ancestor of Penta + Anchi may well have had an embayed parietal, and this might have eventually given rise to the condition seen in Anchi. But the morphological data (ALL of the data, not just a select few frill characters) do not support a direct line of relationship between Pentaceratops and Anchiceratops. This needs to be addressed.
For what it's worth, I do think that the idea that the 'butterfly' ep1s of Anchiceratops resulted from the 'zipping up' of the parietal embayment of something like Pentaceratops is an intriguing idea worth publishing.

Since he has taken up the mantle, I will leave the issues with coding Chasmosaurus epiparietals in the capable hands of fellow reviewer James Campbell.

Since, to my mind, my major concerns still stand, I can't help but stick with my original assessment of 'major revisions' (sorry to be a spoil sport). I'll leave it with our dear editor to decide on the validity of my arguments. If I'm deemed out to lunch, I promise not to throw a fit.

---

## Round 0.3 · accepted · Accept

Please:
-change "Campbell (2014)" to "Campbell (2015)";
-in "Institutional abbreviations", indicate after every institution the city, state [when applicable and even if it is obvious] and country.
All of this can be done while your manuscript is in Production, so I accept it for publication now.
Congratulations again!

·

Basic reporting

See general comments.

Experimental design

See general comments.

Validity of the findings

See general comments.

Additional comments

I thank the authors for incorporating my revisions. The only remaining item I would like them to do is to change "Campbell (2014)" to "Campbell (2015)". This paper (Judiceratops re-evaluation) was first published online in 2014, but was eventually placed into the 2015 volume (volume 52). Please update.

I thank the editor and authors for the opportunity to review this paper. I congratulate the authors on all of their hard work and look forward to the final published version.

Sincerely,

James Campbell